# Near-infrared phosphorescent carbon dots for sonodynamic precision tumor therapy

Bijiang Geng [1], Jinyan Hu[1], Yuan Li[2], Shini Feng[2], Dengyu Pan [1] ✉, Lingyan Feng [3] ✉ & Longxiang Shen [4] ✉

Theranostic sonosensitizers with combined sonodynamic and near infrared (NIR) imaging modes are required for imaging guided sonodynamic therapy (SDT). It is challenging, however, to realize a single material that is simultaneously endowed with both NIR emitting and sonodynamic activities. Herein, we report the design of a class of NIR-emitting sonosensitizers from a NIR phosphorescent carbon dot (CD) material with a narrow bandgap (1.62 eV) and long-lived excited triplet states (11.4 µs), two of which can enhance SDT as thermodynamically and dynamically favorable factors under low-intensity ultrasound irradiation, respectively. The NIR-phosphorescent CDs are identified as bipolar quantum dots containing both p- and n-type surface functionalization regions that can drive spatial separation of $e^- - h^+$ pairs and fast transfer to reaction sites. Importantly, the cancer-specific targeting and high-level intratumor enrichment of the theranostic CDs are achieved by cancer cell membrane encapsulation for precision SDT with complete eradication of solid tumors by single injection and single irradiation. These results will open up a promising approach to engineer phosphorescent materials with long-lived triplet excited states for sonodynamic precision tumor therapy.

Sonodynamic therapy (SDT), which utilizes low-intensity ultrasound (US) and crucial sonosensitizers to generate highly cytotoxic reactive oxygen species (ROS) for inducing tumor cell apoptosis, has recently been highlighted as a promising non-invasive therapeutic modality, because of remarkable therapeutic advantages in extending tissue-penetration deepness and reducing side effects[1-5]. A variety of sonosensitizers have been developed from organic compounds and metal-based inorganic nanoparticles (NPs) for enhanced SDT[4,6-11]. Albeit encouraging progress in the emerging SDT filed, low-yield ROS generation from sonosensitizers presents a major limitation in clinical transformation. It is thus of critical importance to manipulate thermodynamic and dynamical factors that promote the high-yield ROS generation of semiconductor sonosensitizers[12-15]. A narrow bandgap of a sonosensitizer is a beneficial thermodynamic factor because minimal energy is required to drive the generation of electron–hole ($e^- - h^+$)

pairs in low-intensity US settings[6,16-20]. The extended excited-state lifetime of a sonosensitizer is a favorable dynamical factor because long-lived charge carriers can participate in catalytic reactions with high probability before $e^- - h^+$ pair recombination. Based on these considerations, it is a rationale for fabrication of a high-efficacy sonosensitizer that should be endowed with a narrow bandgap and an extended carrier lifetime for effective excitation, separation, and catalytic reactions of charge carriers. Nevertheless, canonical semiconductor sonosensitizers are based on titanium dioxide with a wide bandgap (>3 eV) and a short carrier lifetime (50 ns)[21-23].

With the advent of precision medicine, increasing interest focuses on the development of theranostic nanomedicines for precision cancer therapy[24-26]. Towards this objective, theranostic sonosensitizers with combined sonodynamic and imaging modes are required for imaging guided SDT to determine the optimal SDT window[27,28].

[1]School of Environmental and Chemical Engineering, Shanghai University, Shanghai 200444, China. [2]School of Life Sciences, Shanghai University, Shanghai 200444, China. [3]Materials Genome Institute, Shanghai University, Shanghai 200444, China. [4]Department of Orthopedic Surgery, Shanghai Jiao Tong University affiliated Sixth People's Hospital, Shanghai 200233, China. ✉e-mail: dypan617@shu.edu.cn; lingyanfeng@t.shu.edu.cn; 7250012700@shsmu.edu.cn

Generally, near-infrared (NIR) fluorescent dyes are employed to label sonosensitizers[8,29,30]. However, dye labeling is limited by insufficient photostability, systemic toxicity, and increased fabrication costs. Moreover, typical NIR-fluorescent dyes such as Cy5.5 and indocyanine green (ICG) exhibit no or poor SDT performance owing to rapid $e^-$–$h^+$ recombination[1,4]. Alternatively, NIR-fluorescent semiconductor quantum dots could be stable fluorescence probes, but their time-resolved PL spectra also exhibit high $e^-$–$h^+$ pair recombination rates with a typical carrier lifetime in the range of a few ns[31,32], leading to deactivation of intrinsic sonocatalytic activity. Currently, phosphorescent materials with afterglow emission at room temperature from long-lived excited triplet states have received increasing interest for applications in optoelectronic devices, data security, and time-resolved bioimaging with minimal background interference[33–35]. Notably, electrons in excited triplet states are highly reactive with $O_2$ molecules, a typical triplet-state quencher, to generate highly cytotoxic $^1O_2$[36,37]. As a consequence, phosphorescent materials, in particular, NIR-phosphorescent materials, could be specially designed as enhanced sonosensitizers owing to their ultralong carrier lifetimes and narrow bandgaps. To date, typical NIR-phosphorescent materials have been prepared from transition-metal (Re, Ru, Os, Ir, Pt, Pd, and Au) complexes[38–41], which present limimiations to clinical applications owing to high metal toxicity as well as high costs. In such a scenario, the synthesis of NIR-phosphorescent materials from metal-free materials with excellent biocompatibility and low costs would be a groundbreaking tactic for the exploration of their phosphorescence imaging and SDT performance.

Herein, we report the 'phosphorescent sonosensitizer' concept as a class of theranostic platforms for NIR imaging guided sonodynamic therapy using multifunctional carbon dots (CDs) with an ingenious p–n junction and a greatly narrow bandgap for enhanced charge separation dynamics and NIR excitation/emission as the paragon (Fig. 1). These distinct electronic structures endowed CDs with carrier excitation at low energy, charge separation at high efficiency, and great potential for many modern technologies, including LEDs, solar cells, photocatalytic agents, bioimaging, SDT, and PDT. We have developed a one-step synthetic strategy (microwave synthesis) for controlled synthesis of CDs with the conductivity types of CDs (p–, n–, and p–n junction) facilely engineered by sulfonic acceptors and/or N donors. Compared with previously reported preparation of p–n junction CDs involving complex post-treatments and long-time consumption[42,43], our one-step and high-efficiency method is suitable for low-cost production. Interestingly, we found that the p–n junction CDs (p–n-CDs) possessed bright long-lived (11.4 μs) phosphorescence rather than widely reported short-lived (ns) fluorescence from CDs or graphene quantum dots[42,43]. Although they were dispersed in aqueous solutions or physiological media, they exhibited afterglow emission at room temperature without the need for stabilizing agents such as polyvinyl alcohol (PVA) or other matrices[44–48]. Moreover, they can emit NIR phosphorescence at 760 nm with phosphorescent efficiency as high as 17.6% as a result of the synergistic effects of electron-withdrawing and electron-donating modifications on the electronic structure. We also found that the p–n junction CDs exhibited enhanced $^1O_2$ generation efficiency under US irradiation through three enhancement mechanisms: (1) highly effective inhibition of the $e^-$–$h^+$ pair recombination through the p–n junction, (2) long-lived triplet-state mediated $^1O_2$ generation, (3) GSH depletion using overexpressed GSH as a hole sacrificial agent. Finally, we constructed a CD-based theranostic platform (p–n-CD@CCM) for NIR imaging guided sonodynamic therapy using cancer cell membrane (CCM) as the targeting agent and delivery system to overcome the in vivo limitations by free-standing CDs, such as rapid renal clearance owing to the ultrafine size[49]. Given the NIR imaging capability, excellent biocompatibility, enhanced US-mediated $^1O_2$ generation efficiency of p–n-CDs as well as the CCM targeting capability, we showed complete eradication of solid tumors by single injection and single irradiation. Our results will open

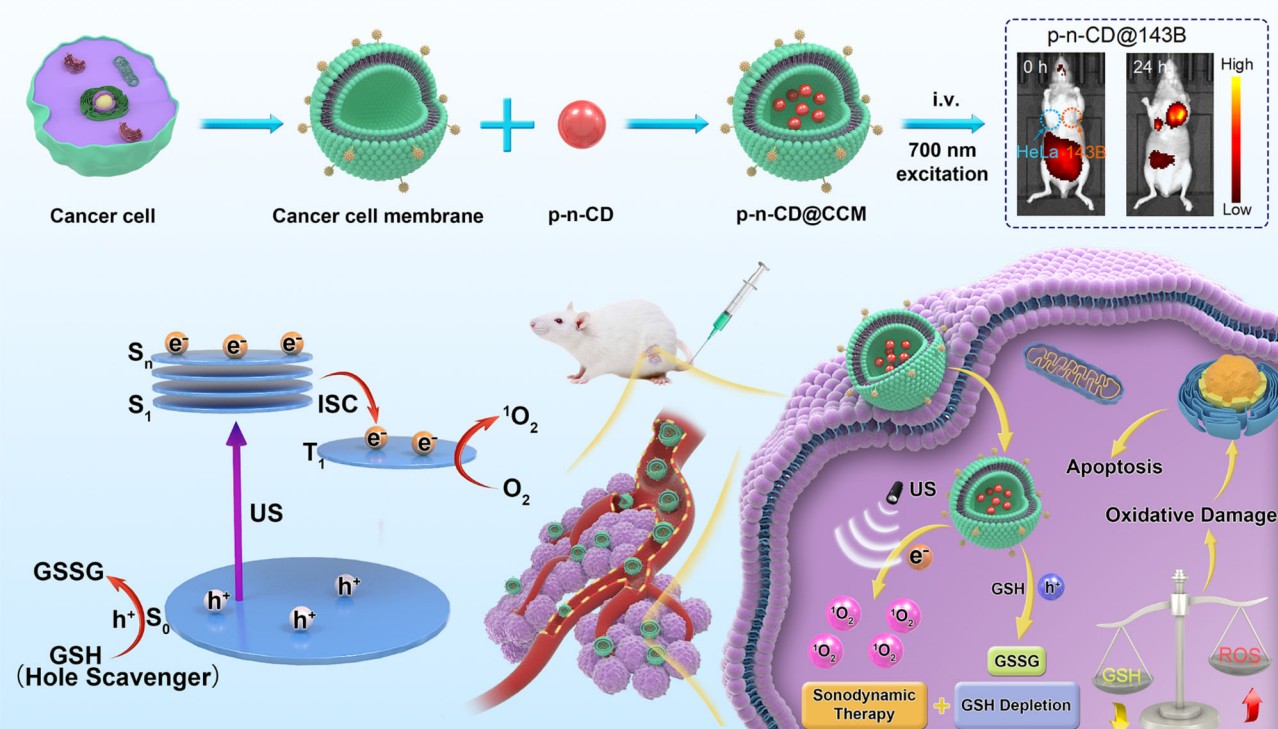

**Fig. 1 | Design and mechanism of p–n-CD@CCM for sonodynamic precision cancer therapy.** Schematic illustration of cancer cell membrane encapsulated NIR-phosphorescent CDs with long-lived triplet excited states ($T_1$) for tumor-specific NIR imaging and precision sonodynamic therapy.

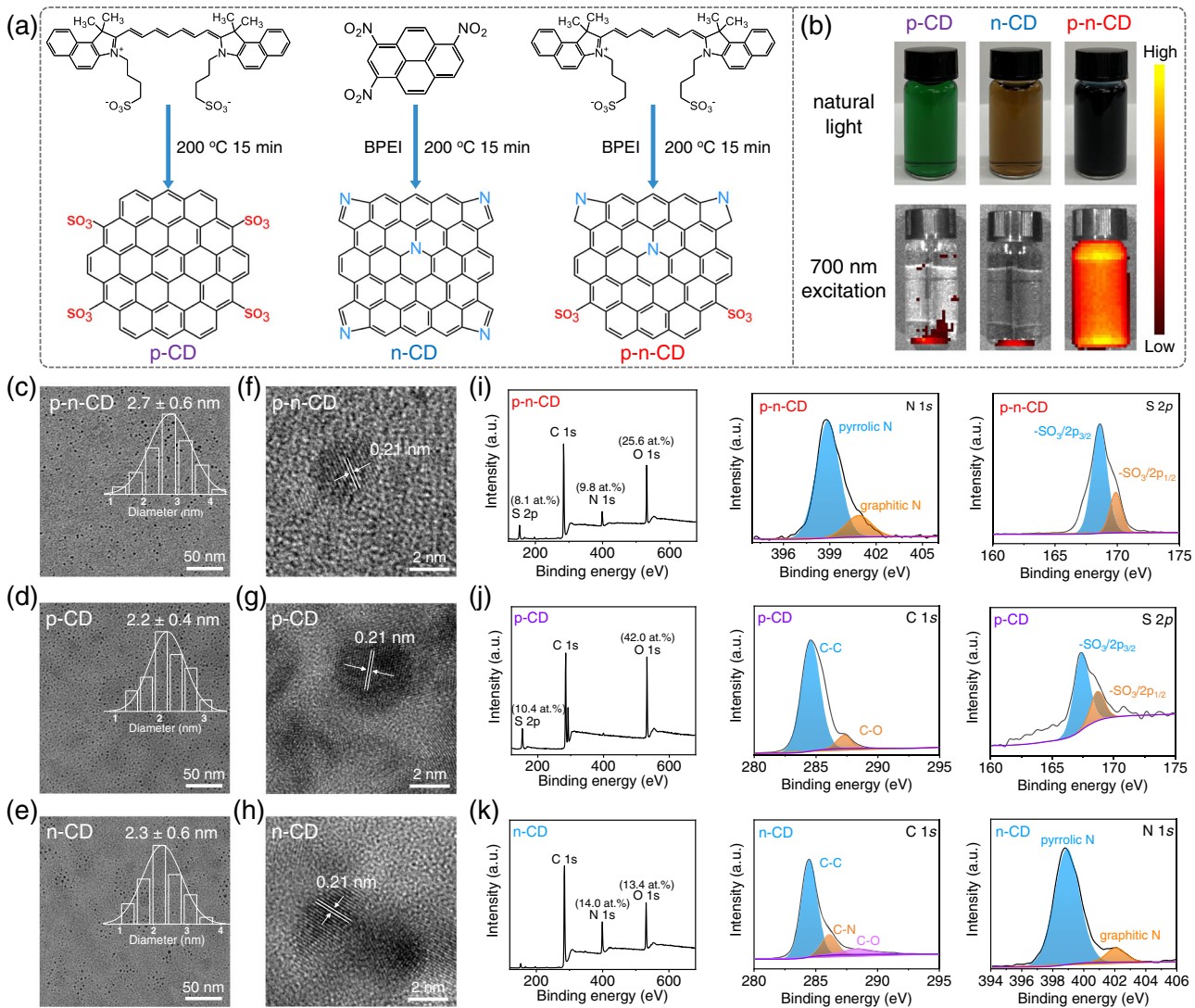

**Fig. 2 | Synthesis and characterization of p-CDs, n-CDs, and p–n-CDs.**
**a** Microwave synthesis routes of p-CDs, n-CDs, or p–n-CDs using ICG, TNP + BPEI, or ICG + BPEI as the precursor, respectively. **b** Photographs and NIR fluorescence images of p-CDs, n-CDs, and p–n-CDs under natural light and 700 nm excitation, respectively. **c–e** TEM images and corresponding lateral size distribution of p–n-CDs (**c**), p-CDs (**d**), and n-CDs (**e**). **f–h** HRTEM images of p–n-CDs (**f**), p-CDs (**g**), and n-CDs (**h**). **i–k** XPS survey, high-resolution C 1s, N 1s, S 2p spectra of p–n-CDs (**i**), p-CDs (**j**), and n-CDs (**k**). A representative image of three biological replicates from each group is shown in (**c–h**). Source data are provided as a Source Data file.

up a promising approach to engineer phosphorescent materials with NIR emission and long-lived triplet excited states for wide-range applications apart from sonodynamic precision tumor therapy.

## Results

### Synthesis of CDs with different acceptor/donator modifications

We developed a one-step microwave strategy for controlled synthesis of NIR-phosphorescent p–n-CDs finely decorated with sulfonic ($SO_3^-$) acceptors and pyrrolic/graphitic N donors from the precursors of ICG and branched polyethylenimine (BPEI) (Fig. 2a). For comparison, p-type CDs (p-CDs) decorated with $SO_3^-$ acceptors and n-type CDs (n-CDs) decorated with pyrrolic/graphitic N donors were synthesized by microwave treatment of ICG only and 1,3,6-trinitropyrene+BPEI, respectively. The resulting p–n-CDs in concentrated aqueous solutions showed black color while concentrated solutions of p-CDs and n-CDs showed green and brown colors, respectively (Fig. 2b). The statistic size distribution from Transmission electron microscope (TEM) images (Fig. 2c–e) showed three kinds of ultrafine CDs have similar average sizes (2.7 ± 0.6 nm for p–n-CDs, 2.2 ± 0.4 nm for p-CDs, 2.3 ± 0.6 nm for n-CDs). Meanwhile, the high crystallinity of these CDs

was observed in their high-resolution TEM (HRTEM) images (Fig. 2f–h), revealing a 0.21 nm crystal spacing corresponded to the (100) plane of graphite[50,51]. A broad distinct peak at approximately 22° can be detected in the X-ray diffraction (XRD) patterns (Supplementary Fig. 1), indicating a wider (002) layer spacing of these CDs[52]. The graphitic structure of the three kinds of CDs was confirmed by the Raman spectra (Supplementary Fig. 2), revealing that the D/G intensity ratio of 0.995, 0.930, and 0.974 for p–n-CDs, p-CDs, and n-CDs, respectively. Their surface modifications were analyzed by X-ray spectroscopy (XPS). The full-range XPS spectrum of p–n-CDs showed a high atomic content of N (9.8 at.%), S (8.1 at.%), and O (25.6 at.%) with S/O ratio approximating to that of $SO_3^-$ groups (Fig. 2i). The high-resolution C 1s XPS spectrum showed peaks at 284.3, 286.0, 291.8 eV, attributable to the C–C, C–N, and C–S bonds, respectively (Supplementary Fig. 3). The high-resolution N 1s spectrum consists of a strong single peak at 399.0 eV and a weak peak at 400.9 eV, which were ascribed to pyrrolic and graphitic N, respectively. The high-resolution S 2p spectrum exhibited two characteristic $SO_3^-$ bands at 168.6 and 169.9 eV. From the XPS analysis, p–n-CDs is characterized by modification with $SO_3^-$ acceptors (8.1 at.%), pyrrolic N (7.3 at.%), and graphitic N (2.5 at.%)

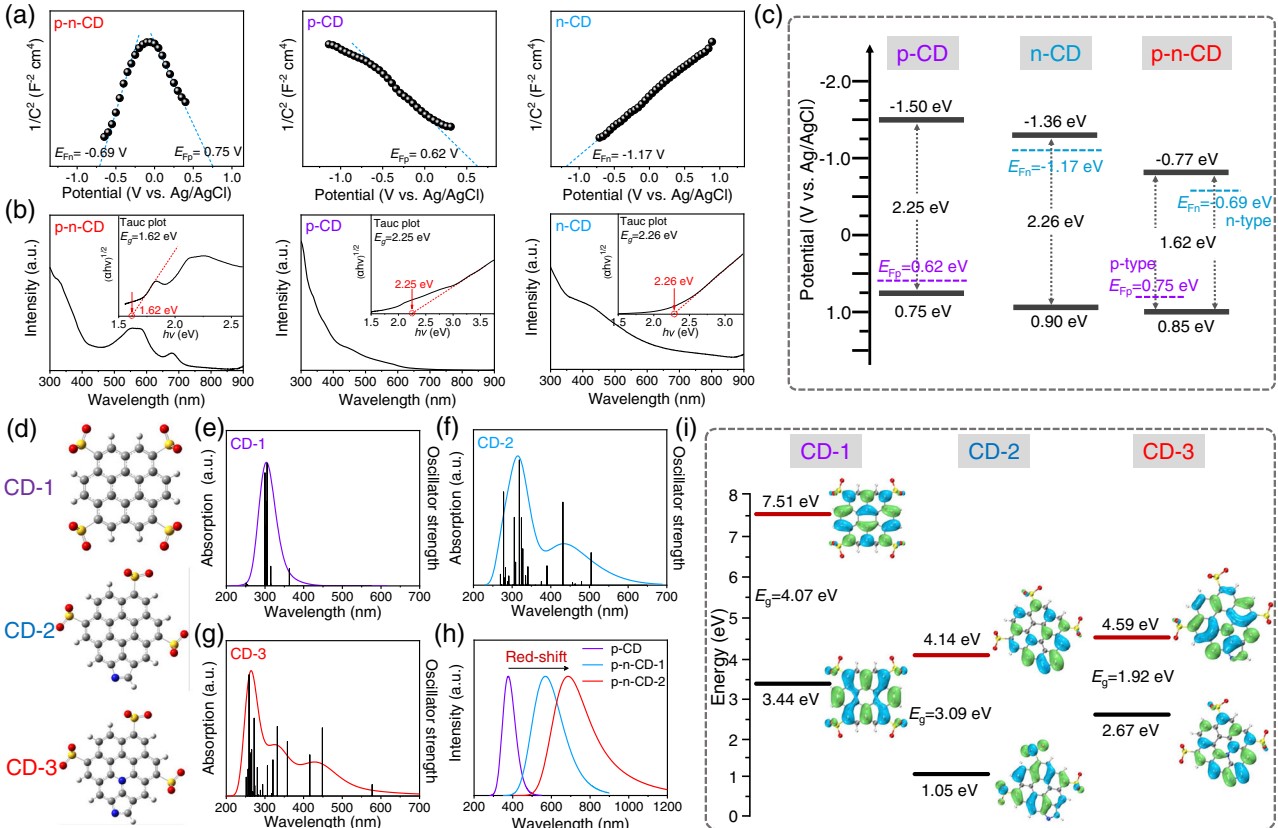

**Fig. 3 | Tuning electronic structures of CDs with different modifications.**
**a** Conductivity type identification of CDs by measuring varied capacitance (*C*) with the applied potential in the Mott-Schottky relationship. The symbols $E_{Fp}$ and $E_{Fn}$ represent the Fermi levels of the p- and n-type semiconductors.
**b** Absorption spectra and corresponding Tauc plots of p−n-CDs, p-CDs, and n-CDs. **c** Schematic illustration of the energy-band diagrams of the p-CDs, n-

CDs, and p−n-CDs. **d** Molecular Models of CDs with different surface modifications (SO₃-modified CD-1, SO₃-pyrrolic N-modified CD-2, and SO₃-pyrrolic N-graphitic N-modified CD-3). **e–i** Calculated UV−vis absorption spectra (**e–g**), fluorescence emission spectra (**h**), HOMO, LUMO, and bandgap (**i**) of CD-1, CD-2, and CD-3 using TD-DFT methods. Source data are provided as a Source Data file.

donators. For p-CDs, the XPS analysis showed their surface modification by rich $SO_3^-$ acceptors (10.4 at.%) (Fig. 2j and Supplementary Fig. 4). For n-CDs, their surface modification was dominated by pyrrolic and graphitic N donators (Fig. 2k and Supplementary Fig. 5). The surface modification of the three types of CDs was further demonstrated by Fourier transform infrared (FT-IR) spectra (Supplementary Fig. 6). The stretching vibration bands of $-SO_3$, C–N, C = C, and C–S can be found in the FT-IR spectrum of p−n-CDs, suggesting the presence of $SO_3^-$ and N-containing heterocycles. Because of negative charge from $SO_3^-$ and positive charge from pyrrolic N, p-CDs, n-CDs and p−n-CDs showed Zeta potential of $−25.3 \pm 3.2$ mV, $+15.6 \pm 2.1$ mV, and $−10.9 \pm 1.6$ mV, respectively (Supplementary Fig. 7). Supplementary Fig. 8 revealed that p−n-CDs were kept stable in physiological conditions (PBS and fetal bovine serum) because of high water solubility, indicating the excellent stability of p−n-CDs.

**Modification-tuned electronic structures of CDs**

We then explored the modification-dependent electronic properties of the CDs with similar sizes. Electrochemical impedance spectroscopic analyses were conducted to identify their conductivity type, as shown in Fig. 3a. Like previously reported p−n-type graphene oxide quantum dots[42], the p−n-CD electrode showed a Mott-Schottky relationship characterized by two straight lines with negative and positive slopes located in different potential ranges, confirming the co-existence of both p- and n-type conductivities. The Mott-Schottky relationship of the p-CD electrode showed a negative slope, corresponding to single p-type conductivity, while the Mott-Schottky relationship of the n-CD electrode exhibited a

positive slope, indicating the n-type conductivity. These results showed that the conductivity types of CDs can be successfully regulated by controlling surface modification with electron-withdrawing and electron-donating moieties.

The bandgap of these CDs was then determined by measuring UV-vis absorption spectra in aqueous solutions (Fig. 3b), from which the $(\alpha E)^2$ versus (E) plots were derived, where E is photon energy and α is the normalized adsorption coefficient. The p−n-CDs showed broad UV to NIR absorption with prominent absorption bands between 500 and 700 nm, suggesting a narrow bandgap. Based on the $(\alpha E)^2−E$ plot, the bandgap of p−n-CDs was determined to be 1.62 eV. In contrast, the absorption spectra of p-CDs and n-CDs were confined in the UV-vis region with an absorption shoulder around 450 nm, and their bandgap was determined as 2.25 and 2.26 eV, respectively. According to electrochemical cyclic voltammetry (CV) measurements on CD electrodes (Supplementary Fig. 9), the conduction band minimum (CBM) of p-CDs, n-CDs and p−n-CDs, was determined to be located at −1.50, −1.36, and −0.77 eV (vs Ag/AgCl), respectively. Accordingly, the valence-band maximum (VBM) of p-CDs, n-CDs, and p−n-CDs was determined to be at 0.75, 0.90, and 0.85 eV (vs Ag/AgCl), respectively, based on VBM = CBM + bandgap. As demonstrated in energy-band diagrams of these CDs (Fig. 3c), the energy-band characteristics of CDs, including CBM, VBM, and the CBM-VBM gap are also dependent on their surface modification. Compared with p-CDs and n-CDs, p−n-CDs have a remarkable reduction in the CBM−VBM gap and much lower CBM position, which can be ascribed to the synergistic effects from simultaneous $SO_3$ and pyrrolic N/graphitic N modification on the shift of the electronic bands.

To understand the synergistic effects of electron-withdrawing and electron-donating modification on CD electronic structures, we established three kinds of $sp^2$ hybrid carbon molecules as simple CD models (Fig. 3d) and evaluated their electronic and optical properties using time-dependent density functional theory (TD-DFT). It should be noted that the relative comparison among the models could identify trends and suggest interpretations for the observed results. Figure 3e-i exhibited the absorption spectra, fluorescence emission spectra, and HOMO-LUMO energy gaps of the molecular CDs. The CD-1 molecule modified with $SO_3$ only revealed a wide bandgap of 4.07 eV and an absorption spectrum confined in the UV range (Fig. 3e). However, the absorption and fluorescence peaks of CD-2 molecules were obviously red-shifted to the visible region after modification with pyrrolic N (Fig. 3f, h). The obtained CD-2 molecule exhibited a lower bandgap of 3.09 eV compared with CD-1 molecule (4.07 eV) (Fig. 3g). Furthermore, the simultaneous modification with $SO_3$, pyrrolic N, and graphitic N in CD-3 results in the smallest HOMO-LUMO gap (1.92 eV) and the most extended absorption and fluorescence emission. Overall, the consistent experimental and calculation results clarified that the strong electron-withdrawing and electron-donating modification could finely tune the energy bands of CDs.

## NIR phosphorescence, sonodynamic and GSH depletion properties of p-n-CD

We investigated the PL properties of CD solutions, whose electronic structures have been modulated by $SO_3$ and/or pyrrolic N/graphitic N. The wide-bandgap p-CDs and n-CDs emitted short-wavelength fluorescence at 430 nm and 470 nm, respectively, under UV excitation (Fig. 4a, b). In contrast, narrow bandgap p-n-CDs emitted NIR emission with PL maximum at 760 nm and maximum excitation at 700 nm (Fig. 4c). The p-n-CDs have absolute PL quantum yield of 17.6% and NIR emission brightness comparable to that of the clinically used ICG. Moreover, the p-n-CDs showed improved stability against photobleaching. The p-n-CDs exhibited no fluorescence attenuation after storage in aqueous solution at room temperature for 7 days, while ICG showed marked fluorescence quenching after 1D storage (Fig. 4d and Supplementary Fig. 10). The strong and stable PL emission of the water-soluble p-n-CDs in the NIR-I biowindow suggests that they can serve as a superior NIR probe for bioimaging.

Since the carrier lifetime of a semiconductor is a key dynamic parameter for a variety of applications, time-resolved PL spectra of CDs was measured. The time-resolved PL spectra of the p-CDs and n-CDs exhibited fast PL decay with an average lifetime of 3.74 ns and 4.82 ns, respectively (Fig. 4e, f), revealing the CDs' common fluorescence nature with short lifetime. Interestingly, the time-resolved PL spectrum of the p-n-CDs excited at 700 nm and detected at 760-nm emission can be fitted by bi-exponential decay function with an average carrier lifetime of 11.4 μs (Fig. 4g), three orders of magnitude longer than that of the p-CDs and n-CDs. The extended carrier lifetime is an indicator of long-lived triplet excited states, from which electron transition to the ground state occurs by phosphorescence at room temperature. To further confirm the phosphorescence nature of p-n-CDs, PVA was used as a matrix to stabilize triplet excited states. The p-n-CD@PVA films exhibited phosphorescent emission at 778 nm and carrier lifetime as long as 185 ms (Fig. 4h, i). Generally, the co-existence of transient and long-lived components was commonly observed in phosphorescent materials, such as phosphorescent CDs[44,53–55]. To our surprise, there were no transient components (ns-scale lifetime) from the time-resolved PL spectra of the p-n-CD solution and the p-n-CD@PVA film apart from the long-lived components (1.6 and 19.2 μs for the p-n-CD solution; 15.7, 87.6, 637.2 ms for the p-n-CD film). These results suggest that the $e^--h^+$ pair recombination from singlet excited states to the ground state could be inhibited by the p-n junction between different functional regions within single CDs.

Inspired by the narrow bandgap, long carrier lifetime, and p-n junction configuration of NIR-phosphorescent p-n-CDs, we investigated their sonodynamic properties under low-intensity US irradiation (50 kHz, 3.0 W/cm$^2$) using 1,3-diphenylisobenzofuran (DPBF) as the $^1O_2$ probe. With increasing irradiation time from 0 to 5 min, the characteristic absorption peaks of DPBF rapidly disappeared (Fig. 4j), indicating that US-excited p-n-CDs as a sonosensitizer reacted with $O_2$ to produce $^1O_2$ as highly cytotoxic ROS for SDT. In contrast, p-CDs and n-CDs with short carrier lifetime and wide bandgap showed no sonodynamic effect (Supplementary Fig. 11). For comparison with ICG with a narrow bandgap but with a short carrier lifetime (0.46 ns) (Supplementary Fig. 12), p-n-CDs showed notably enhanced US triggered $^1O_2$ generation. The $^1O_2$ generation efficiency was thus determined to be 0.220 min$^{-1}$ for p-n-CDs, 2.2 and 3.2 times that of ICG and $TiO_2$, respectively (Fig. 4k). No significant decrease of DPBF absorption peaks can be observed in the DPBF + US, p-n-CDs + US (no DPBF), and ICG + US (no DPBF) groups (Supplementary Fig. 13), suggesting the US-activated $^1O_2$ generation behavior of p-n-CDs. To present direct evidence for the enhanced ROS generation capability for the p-n-CD sonosensitizer, electron spin resonance (ESR) spectra were measured using 2,2,6,6-tetramethylpiperidine (TEMP) as the trapping agent of $^1O_2$. Compared with ICG and $TiO_2$ sonosensitizers, p-n-CDs exhibited the strongest ESR signals characteristic of $^1O_2$ (Fig. 4l). For non-SDT-active p-CDs and n-CDs, no $^1O_2$ ESR signals were observed. These results suggest that the narrow bandgap and long carrier lifetime are two key factors for activating and enhancing SDT. Apart from the enhanced sonodynamic performance of p-n-CDs, the long-lived triplet excited states could endow p-n-CD with excellent photodynamic properties. As depicted in Supplementary Fig. 14, the characteristic absorption peak of the $^1O_2$ probe DPBF at 412 nm decreased significantly under 660 nm laser irradiation in the presence of p-n-CDs, confirming the high photodynamic activity of p-n-CDs owing to the presence of long-lived triplet excited states. Because of the catalytic activity of long-lived triplet excited states, we proposed a triplet-state mediated $^1O_2$ generation mechanism (Fig. 4p), distinct from that of extensively reported inorganic sonosensitizers[16,56–58]. The long-lived triplet excited states of p-n-CDs could be available through US or NIR light excitation and then intersystem crossing (ISC) for phosphorescence emission and generation of $^1O_2$ because $O_2$ molecules are a typical triplet-state quencher[36,37]. Given the tumor-specific microenvironment (TME) characterized by overexpressed GSH, we further explored the GSH depletion ability of the p-n-CD sonosensitizer using 5,5-dithiobis(2-nitrobenzoic acid) (DTNB) probe. Figure 4m revealed that the characteristic absorption peak of GSH significantly decreased with the extended US time, indicating that GSH could be acted as a hole-scavenger in the SDT process. The sonodynamic effect of p-n-CDs in the presence of GSH was then measured to confirm the enhanced $^1O_2$ generation. It was found that the $^1O_2$ generation capability was further enhanced by GSH, which functions as a hole sacrificial agent to further inhibit $e^--h^+$ pair recombination (Fig. 4n, o).

## In vitro precision SDT of p-n-CD@CCM

It is known that typical inorganic nanomedicines with suitable sizes (20–200 nm) could circumvent rapid renal filtration and enable passive accumulation in tumor tissue by the enhanced permeability and retention (EPR) effect[59–62], but these nanomedicines would be limited in clinical translation because of long-term safety concerns. Instead, recent studies have revealed that ultrasmall CDs could be used as highly biocompatible nanomedicines because they undergo rapid renal filtration without inducing significant long-term toxicity[49]. A critical issue to be addressed is how to improve tumor targeting or accumulation levels and elongate blood circulation half-life for CD-based nanomedicines.

Recently, cancer cell membranes (CCM) derived from homologous tumors have been developed as efficient tumor-specific

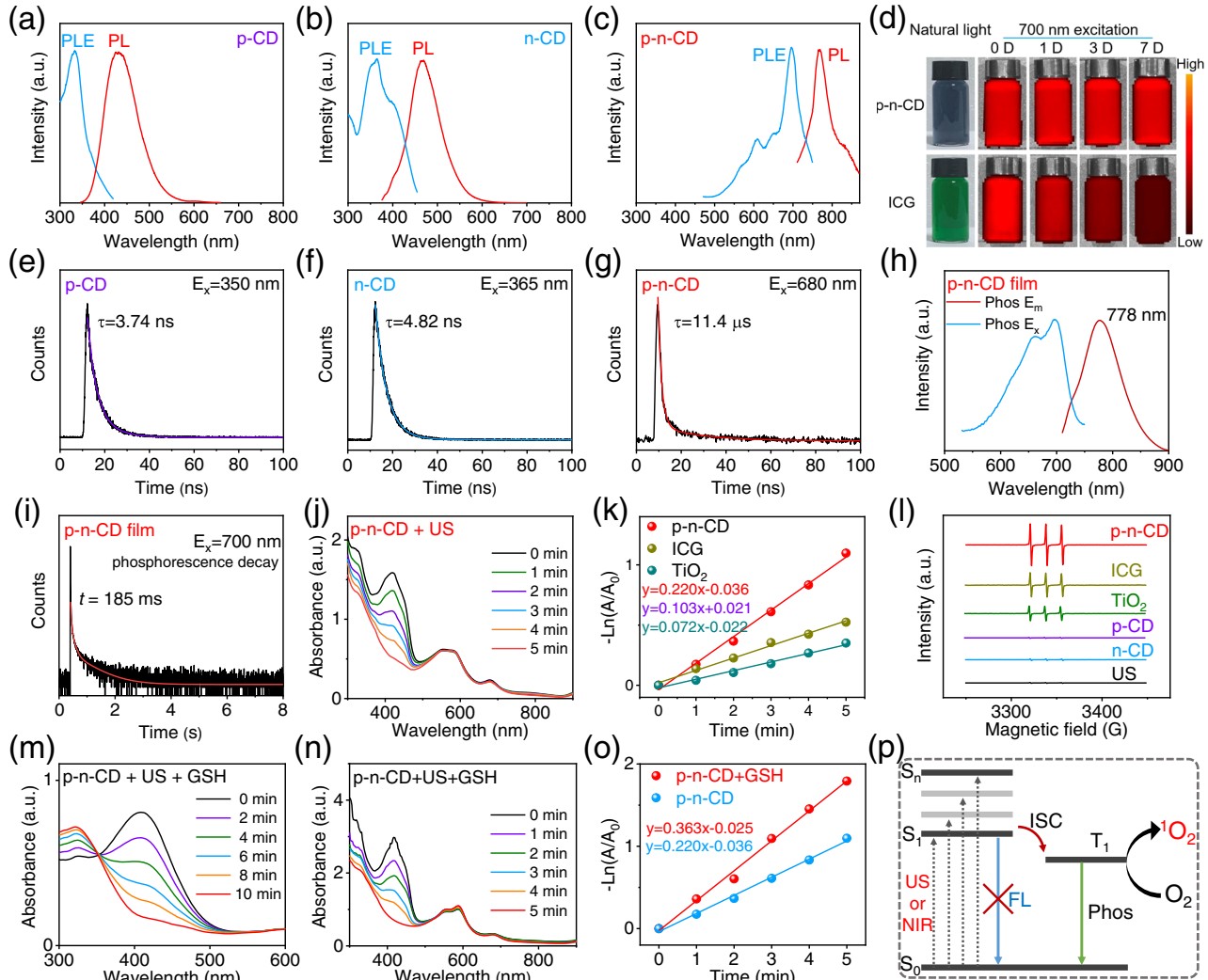

**Fig. 4 | NIR phosphorescence, sonodynamic and GSH depletion properties of p−n-CDs. a–c** PL and excitation (PLE) spectra of p-CDs (**a**), n-CDs (**b**), and p−n-CDs (**c**). **d** NIR emitting stability contrast between p−n-CDs and ICG in aqueous solutions stored for 0, 1, 3, and 7 days. The excitation and emission wavelengths were 700 and 790 nm, respectively. **e–g** Time-resolved PL spectra and the corresponding fitting curves of p-CDs (**e**), n-CDs (**f**), and p−n-CDs (**g**). **h** Phosphorescence excitation and emission spectra of p−n-CD@PVA films. **i** Time-resolved phosphorescence spectrum and the corresponding fitting curve of p−n-CD@PVA. **j** Time-dependent $^1O_2$ generation of p−n-CDs under US irradiation (50 kHz, 3.0 W/cm²) for 5 min detected with the DPBF probe. **k** Rate constant of US triggered $^1O_2$ generation in the presence of p−n-CDs, ICG, and TiO₂ under the same US irradiation. **l** ESR spectra of US triggered $^1O_2$ generation in the absence and presence of p−n-CDs, p-CDs, n-CDs, ICG, or TiO₂ using TEMP as the trapping agent of $^1O_2$. **m** GSH depletion by holes produced from US-excited p−n-CDs. **n** Enhanced US triggered $^1O_2$ generation by p−n-CDs in the presence of GSH (1 mM) as the hole quencher. **o** Rate constant of $^1O_2$ generation in the presence of p−n-CD under US irradiation with or without the addition of GSH (1 mM). **p** Schematic illustration of phosphorescence emission and $^1O_2$ generation mechanisms of p−n-CDs from long-lived excited triplet states via ISC. The concentrations of indicated samples (p−n-CD, p-CD, n-CD, ICG, and TiO₂) were all 200 µg/mL. Source data are provided as a Source Data file.

delivery vehicles because cancer-specific proteins are retained on the surface enabling homotypic recognition to the same cancer cell lines[63–65]. Encouraged by the promise of CCMs in cancer-specific drug delivery, we prepared two kinds of p−n-CD loaded CCM vehicles, p−n-CD@143B and p−n-CD@Hela, derived from human osteosarcoma 143B cells and Human cervical cancer Hela cells, respectively, by a successive extrusion approach.

TEM images of representative p−n-CD@CCM vehicles revealed the superficial coverage of the CCM layer and the presence of loaded p−n-CDs in the interior cavity, indicating that ultrafine p−n-CDs were successfully encapsulated into CCM vehicles (Fig. 5a, b and Supplementary Fig. 15). The amount of encapsulated p−n-CDs in p−n-CD@CCM was determined by UV-vis-NIR absorption spectroscopy. After removing unloaded p−n-CDs through centrifugation, the loading rate of p−n-CD in p−n-CD@CCM can be determined to be 20%. Western blotting analysis (Fig. 5c) showed source membrane markers were

well retained in the CCM vehicles, including two plasma membrane-specific markers (cadherins and Na+ /K+ -ATPase) and a widely reported tumor-associated transmembrane protein (glycoprotein 100 or gp100). In contrast, intracellular protein markers of Histone H3 and COXIV were eliminated from CCMs, demonstrating the selective retention of membrane fragments after the treatment. The average hydrodynamic diameter of unloaded CCM and p−n-CD@CCM vehicles was $68.1 ± 1.9$ nm and $78.8 ± 1.3$ nm, respectively (Fig. 5d). No visible precipitate was observed in PBS and FBS solutions after storing for 7 days (Supplementary Fig. 17a), demonstrating the excellent stability of p−n-CD@CCM in the physiological environment for medical applications. The high stability of p−n-CD@CCM in aqueous solution was further evaluated by measurement of change in hydrodynamic diameter and Zeta potential during storage. There was no substantial change in hydrodynamic diameter and Zeta potential after storing for 7 days (Supplementary Fig. 17b, c).

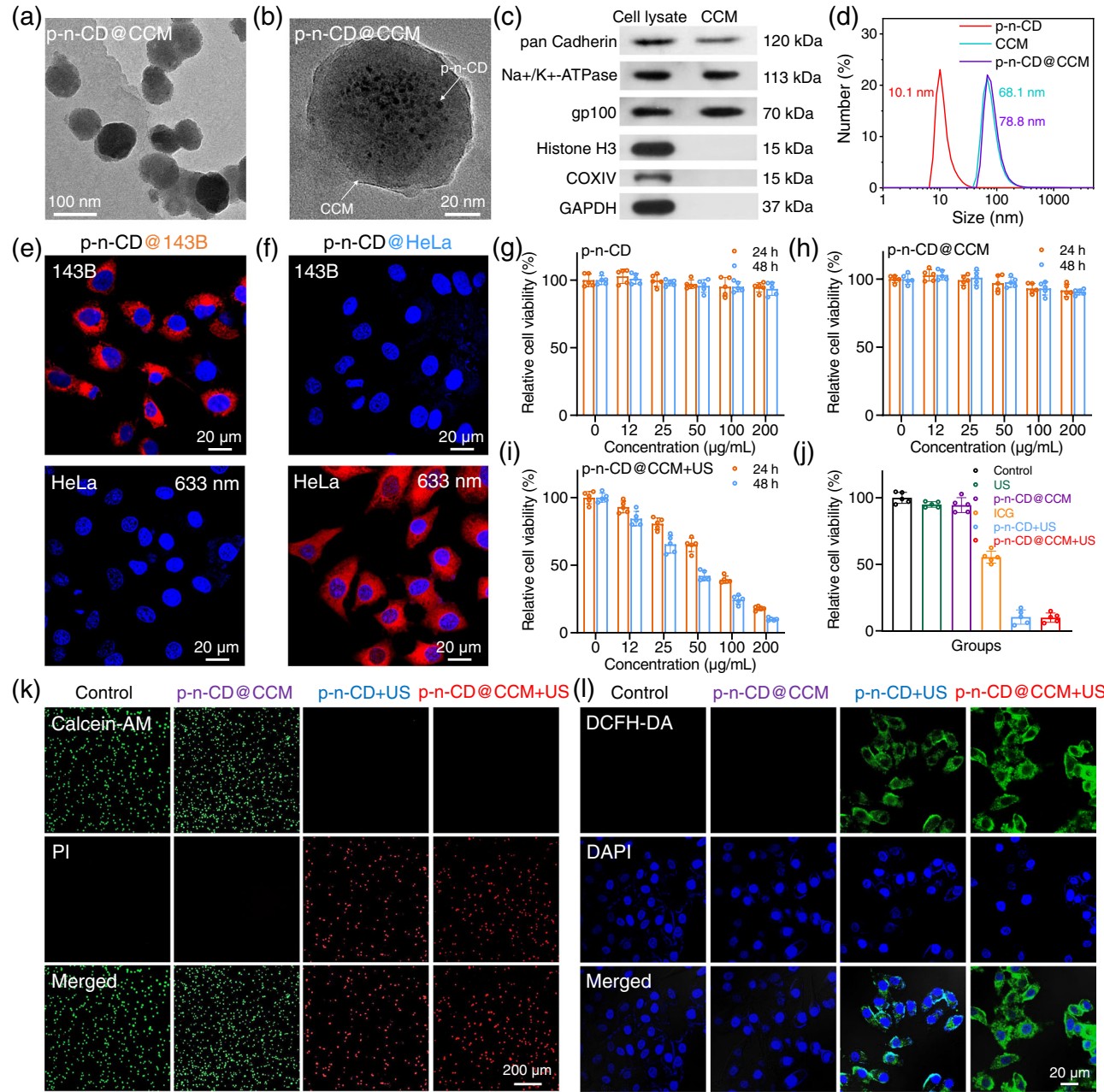

**Fig. 5 | In vitro precision SDT of p−n-CD@CCM. a, b** TEM images of p−n-CD@CCM. **c** Western blotting analysis of cell lysate and CCM by membrane-specific (pan-cadherin, Na+/K+-ATPase, and gp100) and intracellular protein markers (Histone H3 and COXIV). **d** Hydrodynamic diameter of free p−n-CDs, CCM, and p−n-CD@CCM. **e, f** Cancer-specific confocal images of 143B and Hela cells incubated with p−n-CD@Hela or p−n-CD@143B, respectively. **g, h** Relative cell viability of 143B cells incubated with free p−n-CDs (**g**) or p−n-CD@CCM (**h**) at varied concentrations without US irradiation (n = 5 biologically independent samples). **i** Relative cell viability of 143B cells incubated with p−n-CD@CCM under US irradiation (50 kHz, 3.0 W/cm²) (n = 5 biologically independent samples). **j**–**l** Relative cell viability (**j**) (n = 5 biologically independent samples), live/dead cell staining (**k**), and ROS staining (**l**) of 143B cells after indicated treatments. Data are presented as mean values ± SD. A representative image of three biological replicates from each group is shown in (**a**–**c**), (**e**), (**f**), (**k**), (**l**). Source data are provided as a Source Data file.

We then investigated the sonodynamic properties of p−n-CD@CCM, revealing that the characteristic absorption peaks of DPBF at 412 nm rapidly disappeared with increasing the US irradiation time (Supplementary Fig. 18a). The ¹O₂ generation efficiency of p−n-CD@CCM was thus calculated to be 0.212 min⁻¹ (Supplementary Fig. 18b), which was similar to that of pristine p−n-CD (0.220 min⁻¹), indicating that encapsulation into CCM will not affect the excellent sonodynamic performance of p−n-CDs. To confirm the homotypic recognition capability and assess in vitro targeting efficacy of p−n-CD@CCM vehicles, p−n-CD@143B was incubated with 143B cells using Hela cells as a negative control, and

cell images excited at 633 nm were taken by a confocal laser scanning microscopy (CLSM). The sharp contrast between bright green fluorescence of 143B cells and dark green fluorescence of Hela cells revealed that p−n-CD@143B can specifically recognize 143B cells with high uptake levels owing to the retained 143B cell membrane markers (Fig. 5e). Similarly, p−n-CD@Hela can specifically differentiate Hela cells from 143B cells, as shown in Fig. 5f. These results clearly demonstrated it is feasible to design p−n-CD@CCM as the cancer-specific NIR probe for precise bioimaging via CCM-based homotypic recognition. The in vitro imaging also showed that p−n-CDs as a sonosensitizer can be facilely delivered

by CCM vehicles and internalized into the cytoplasm of target cells for SDT.

Prior to in vitro SDT evaluation, the biosafety of p−n-CD@CCM at a cellular level was assessed. The cytotoxicity of p−n-CDs and p−n-CD@CCM was tested by the MTT assay at varied concentration of 0–200 μg/mL. It was found that no obvious cytotoxicity was induced by p−n-CDs or p−n-CD@CCM even at 200 μg/mL, suggesting a good biosafety for SDT application (Fig. 5g, h). To reveal the in vitro SDT performance of p−n-CD@143B against target 143B cells, we performed the MTT assay under US irradiation (50 kHz, 3.0 W/cm$^2$) for 5 min. Substantial concentration-dependent cell toxicity was induced by p−n-CD@CCM even at lower concentrations under US irradiation (Fig. 5i). To demonstrate the enhanced SDT performance of p−n-CD@CCM, relative cell viability of 143B cells against the control for 48 h incubation was compared for different treatment groups, including US, p−n-CD@CCM, ICG + US, p−n-CD + US, and p−n-CD@CCM + US under the same concentration (200 μg/mL) and US irradiation conditions. Both p−n-CD + US and p−n-CD@CCM + US groups exhibited the highest SDT effect with little discrepancy at the cellular levels (Fig. 5j). It should be pointed out that ICG has the same functions for NIR imaging and SDT as p−n-CDs, but it only exhibited a lower SDT effect, because of its much shorter carrier life than that of p−n-CDs.

We further investigated the SDT effect of p−n-CD@CCM using the calcein-AM and PI co-staining assay. Remarkable cell death corresponding to PI-positive cells was observed by fluorescence microscope after p−n-CD mediated SDT or p−n-CD@CCM mediated SDT (Fig. 5k and Supplementary Fig. 19, 21a), confirming the excellent SDT efficacy of p−n-CDs or p−n-CD@CCM. Moreover, we performed the apoptosis assay through the flow cytometry using Annexin V-FITC and PI staining. As presented in Supplementary Fig. 22, significant apoptosis of 143B cells was detected in the p−n-CD + US and p−n-CD@CCM + US groups. To investigate the antitumor mechanism of p−n-CDs in cellular level, we evaluated the intracellular ROS levels using DCFH-DA staining assay. The 143B cells in the control group, US irradiation alone group, and p−n-CD@CCM alone group exhibited no obvious ROS fluorescence signal (Fig. 5l and Supplementary Figs. 20, 21b). In contrast, the strongest green fluorescence located in the cytoplasm was observed in p−n-CD + US and p−n-CD@CCM + US groups, confirming the ROS-mediated cell death induced by p−n-CD-based SDT.

## In vivo precision SDT of p−n-CD@CCM

To achieve precision SDT of tumors, we first assessed the tumor-specific in vivo NIR imaging capability of p−n-CD@143B. 143B and Hela tumors were simultaneously transplanted at two sides of the same mice for in vivo NIR imaging with intravenously injected free p−n-CDs and p−n-CD@143B as the probe excited at 700 nm. For free p−n-CDs, there was no substantial accumulation at 143B or Hela tumor sites because of rapid renal filtration of p−n-CDs (Fig. 6b and Supplementary Fig. 23a). Interestingly, p−n-CD@143B specifically accumulated in 143B tumors because of the homotypic recognition nature of CCM-143B (Fig. 6c and Supplementary Fig. 23b). The highest NIR fluorescence signal in the tumor tissue was detectable at 12-24 h post-injection. Significant retention of p−n-CD@143B was mainly found in the liver and the 143B tumor at the 12 h and 24 h post-injection, with little retention in the Hela tumor (Fig. 6d and Supplementary Fig. 24a). Similarly, p−n-CD@Hela can specifically accumulated in Hela tumors, as depicted in Supplementary Figs. 24, 25, 26b. These results clearly verified the excellent in vivo imaging capability of p−n-CD@CCM at the NIR-I biowindow, tumor-specific recognition potential, and high intratumoral levels.

We next evaluated in vivo SDT effects of p−n-CD@CCM for precision tumor therapy. Six groups of 143B tumor-bearing mice were randomly divided as followed: Group 1: Control, Group 2: US, Group 3: p−n-CD@CCM, Group 4: ICG + US, Group 5: p−n-CD + US, Group 6: p−n-CD@CCM + US. According to the in vivo NIR imaging result, single

US irradiation (50 kHz, 3.0 W/cm$^2$, 5 min) was performed at 24 h after single i.v. injection of p−n-CD@CCM (2.0 mg/kg) (Fig. 6a). The time-dependent change of treated 143B tumors in volume and weight exhibited consistent SDT effects (Fig. 6e, f). That is, no tumor growth inhibition effects were observed in the US, ICG + US, and p−n-CD@CCM groups; a limited tumor inhibition was induced by p−n-CD + US; a complete tumor eradication was realized by p−n-CD@CCM + US during 16 days (Supplementary Fig. 27). Importantly, the mice percent survival was greatly improved by p−n-CD@CCM + US, with no death observed in the 60-day treatment/observation duration (Fig. 6g). The excellent SDT therapeutic effects of p−n-CD@CCM were also demonstrated by the ROS staining of tumor tissues. The p−n-CD@CCM + US group possessed the highest ROS levels in tumor tissues (Fig. 6i). Haematoxylin and eosin (H&E) staining was then carried out to analyze the SDT efficacy of p−n-CD@CCM, which exhibited the most severe damage of tumors (Fig. 6j). To investigate whether tumor cells in tissue were apoptosis on day 4 or 5, the tumor sections in each groups were collected on the fourth day post-treatment and subjected to terminal deoxynucleotidyl transferase dUTP nick-end labeling (TUNEL) staining. Supplementary Fig. 28 revealed that the tumor in p−n-CD@CCM + US group showed severe apoptosis and necrosis. Together, these in vivo SDT effects further revealed that it is of importance to design a high-efficacy sonosensitizer with tumor-specific targeting capability, as confirmed by the model p−n-CD@CCM sonosensitizer.

Considering that US wave possessed high tissue-penetration depth, the in vivo deep-tissue SDT was then performed to evaluate the potential therapeutic effect of p−n-CD@CCM. We established two 143B tumors on the left and right sides of mice and the US treatments were performed from left to right (Supplementary Fig. 29a). The left and right tumors rapidly grew in the control group (Supplementary Fig. 29b). Significantly, the tumor growth on both sides was completely inhibited for mice treatment with p−n-CD@CCM plus US irradiation from left to right. These results clearly demonstrated that the US treatments could penetrate through the body of mice and the high-efficiency p−n-CD@CCM sonosensitizers could be applicable for deep-tissue SDT.

The pharmacokinetic examination of p−n-CD and p−n-CD@CCM sonosensitizers after systemic injection were then investigated. The plasma concentration-time profiles of p−n-CD and p−n-CD@CCM samples showed a typical double-compartment pharmacokinetic behavior (Supplementary Fig. 30). For free p−n-CDs without CCM coating, the elimination half-life (t$_{1/2}$) was only $0.8 \pm 0.1$ h, revealing rapid renal clearance owing to the ultrafine size ($2.2 \pm 0.4$ nm), like previous reports on CDs. In contrast, after CCM coating, the half-life was elongated to $11.6 \pm 0.9$ h, 14.5 times of that of free p−n-CDs, indicating that the typical renal clearing process of CDs was inhibited by CCM coating. The longer circulation time allowed p−n-CD@CCM to enrich in tumors at high levels. The in vivo biodistribution and excretion pathway of p−n-CD@CCM were explored for examination of its in vivo metabolic process. Supplementary Fig. 31a revealed that p−n-CD@CCM primarily accumulated in liver at 24 h post-injection due to the capture of the reticuloendothelial system. Importantly, no obvious p−n-CD signals were observed in these organs after 14 days, suggesting the nearly complete clearance of p−n-CD@CCM. To study the excretion pathway, the p−n-CD concentrations in the urine and feces were also measured. Much higher levels of p−n-CD were observed in the feces, demonstrating that p−n-CD@CCM was eliminated from the mice majorly by liver excretion (Supplementary Fig. 31b).

Finally, we investigated the potential long-term toxicity of p−n-CD@CCM through the histopathological measurement, biochemical blood analysis, and blood routine examination. During the treatment period, negligible fluctuation of the mice weight can be observed in each group (Fig. 6h). For histopathological measurement, no obvious damages in major organs were detected after different treatments

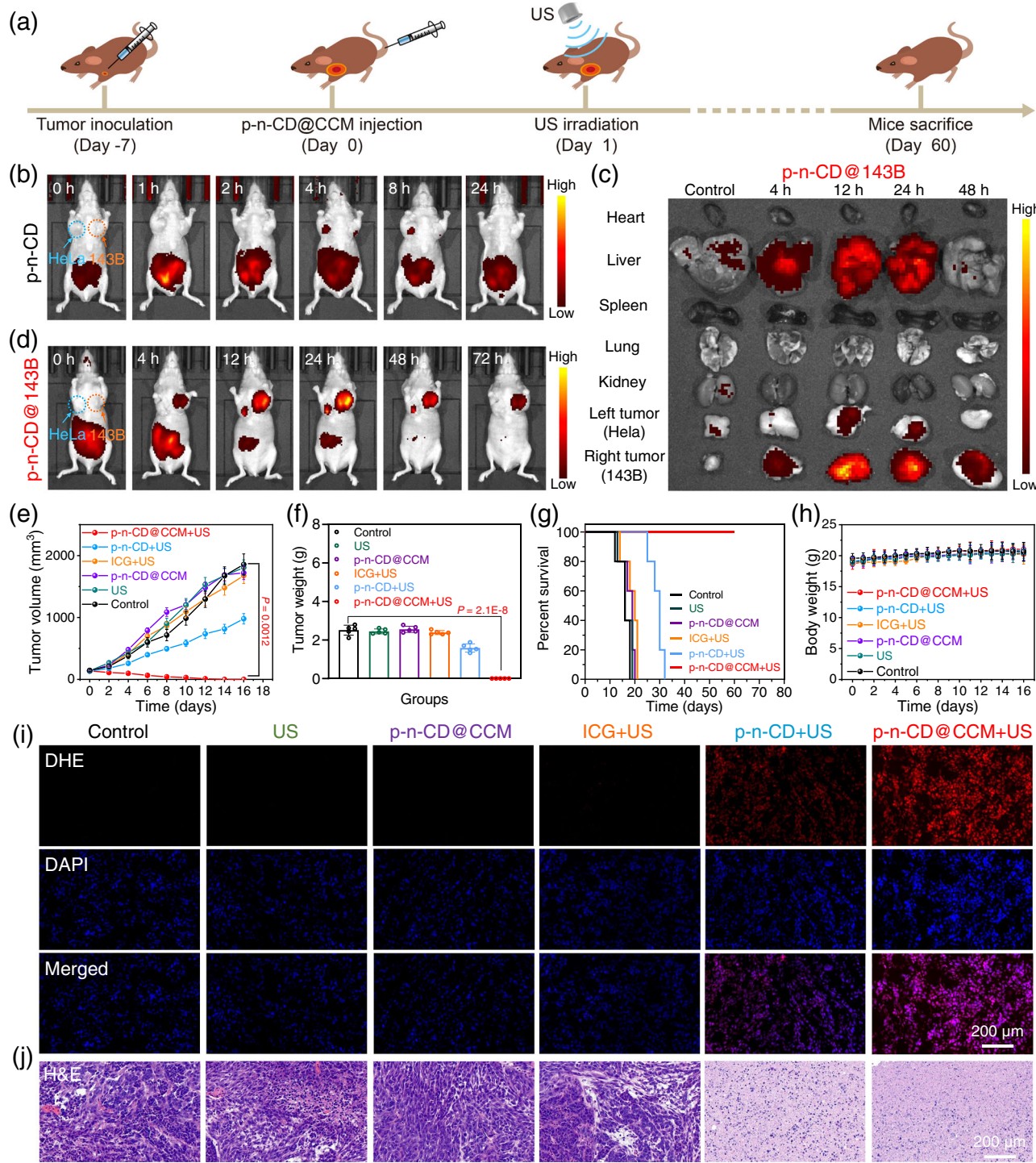

**Fig. 6 | In vivo precision SDT of p−n-CD@CCM. a** SDT protocol of p−n-CD@143B (2.0 g/kg) under single US irradiation on 143B tumor-bearing mice. **b** In vivo NIR imaging of mice with free p−n-CDs. **c** Tumor-specific in vivo NIR imaging of mice with p−n-CD@143B. **d** Ex vivo NIR imaging of major tissues and tumors with p−n-CD@143B. **e** 143B tumor growth curves of mice after indicated treatments (*n* = 5 biologically independent samples). **f** 143B tumor weight after indicated treatments (*n* = 5 biologically independent samples). **g** Survival curves of mice after indicated treatments. **h** Body weight of mice after indicated treatments. **i** ROS staining images of tumors after indicated treatments with DHE as the ROS probe. **j** H&E staining images of tumors after indicated treatments. Data are presented as mean values ± SD. Statistical significance was calculated with two-tailed Student's *t* test (**e**) and (**f**). A representative image of three biological replicates from each group is shown in (**i**), (**j**). Source data are provided as a Source Data file.

(Supplementary Fig. 32), indicating the insignificant adverse effects of p−n-CD@CCM. Furthermore, all the blood biochemical and blood routine indexes in each groups were located in the normal range (Supplementary Fig. 33), suggesting the good biocompatibility of p−n-CD@CCM.

## Discussion

Recently, CDs have been engineered as promising fluorescent materials owing to their numerous merits such as high optical stability, excellent solubility, low toxicity, and easy preparation. Generally, single-component CDs do not exhibit observable phosphorescence

due to their unstable triplet states. To extend their phosphorescent applications in data encryption and anti-counterfeiting, great effort has been made in the construction of CD-based hybrid phosphorescent systems, in which various organic or inorganic matrices, including KAl$(SO_4)_2 \cdot x H_2O$[45], $SiO_2$[33], zeolites[46], layered double hydroxides[47], poly(vinyl alcohol)[44], polyurethane[66], and cyanuric acid[48], were employed to suppress the intermolecular vibration for triplet stabilization. However, CD-based hybrid phosphorescent systems are limited by the solid-state-only applications as well as time-consuming and high-cost fabrication involving the use of additional matrices in the solid composites. To date, matrix-free phosphorescent CDs have not been fabricated for phosphorescent applications in aqueous solutions or physiological media because several issues are difficult to address by self-stabilization of triplet states. First, a challenge is to substantially reduce the energy gap ($\Delta E_{ST}$) of CDs between the lowest singlet (S1) to the lowest triplet (T1) for enhancing their favorable intersystem crossing. Second, highly efficient fluorescent emission in CD solutions shows overwhelming advantages over phosphorescent emission, leading to low population probability in the triplet states despite effective ISC. Unfortunately, the electron–hole recombination from S1 to the ground state could not be inhibited to quench the fluorescence emission for enhanced RTP. Third, the excited triplet states at a low-level population are strongly quenched by dissolved molecular oxygen (triplet-state $^3O_2$) to form singlet-state $^1O_2$, further impeding the aqueous applications of RTP materials. Herein, we reported that matrix-free p–n-CDs in solution states can be rationally constructed as a solution-applicable RTP material by p–n junction engineering. The formation of the p–n junction within single p–n-CDs can effectively inhibit the e$^-$–h$^+$ pair recombination from excited singlet states to the ground state and thus enhance the RTP emission as evidenced by the time-resolved PL spectra of the p–n-CD solution and the p–n-CD@PVA film, where no transient fluorescent components were observed apart from the long-lived components. In addition, rapid charge transfers from the n-type dopant (pyrrolic N) to the p-type dopant ($SO_3^-$) in the bipolar configuration could stabilize the triplet states with a long lifetime of 11.4 μs in aqueous solutions. In contrast, for p-CDs with $SO_3^-$ functionalization only or n-CDs with pyrrolic/graphitic N functionalization only, effective charge transfer could not be established, leading to fluorescent emission with a ns-scale lifetime in aqueous solutions. Furthermore, TD-DFT calculations showed that the simultaneous modification with $SO_3$, pyrrolic N, and graphitic N in the CD-3 model can reduce $\Delta E_{ST}$ to a smaller value (0.55 eV), which is favorable for effective ISC to populate triplet excitons and consequently emitting phosphorescence. In contrast, for $SO_3$-modified CD-1 has $\Delta E_{ST}$ as high as 1.08 eV, suggesting ineffective ISC at room temperature. These results clearly revealed that phosphorescent p–n-CDs exhibited a distinct triplet stabilization mechanism by creating the p–n junction rather than the hydrogen-bonded network in CD-based hybrid phosphorescent systems. Compared with the complex preparation processes of CD-based hybrid phosphorescent systems, the construction of highly phosphorescent p–n junction CDs is simple, one-step, and rapid by one-pot microwave irradiation.

It is noted that current afterglow CDs in the hybrid systems are mainly limited to visible RTP emissions from blue, green to yellow. Recently, high-efficiency red RTP emissions at 620 nm have been also achieved by embedding CDs in Mn-containing open-frame-work matrices[54], but RTP emissions extended to the NIR region remain unexplored to date for NIR applications such as NIR imaging and emitters. Generally, the NIR emissions of quantum dots require a narrow bandgap; it is challenging for CDs to simultaneously reduce the bandgap and stabilize triplet states. Here, we found that phosphorescent p–n-CDs have a narrow bandgap of 1.62 eV and thus emit NIR phosphorescence at 760 nm as a result of the synergistic effects of electron-withdrawing and electron-donating modifications on the p–n-CD electronic structure. The rational p–n junction modification in NIR-

phosphorescent p–n-CDs creates the narrow bandgap and stabilize triplet states. Importantly, the p–n-CDs exhibit a phosphorescent efficiency as high as 17.6% in aqueous solutions and high NIR emission brightness comparable to that of the clinically used ICG. These RTP properties render p–n-CDs as a superior long-lived NIR imaging platform for biomedical applications with minimal background interference.

Long-lived triplet states of organic photosensitizers are tremendously important for photodynamic therapy (PDT) owing to a high yield of singlet oxygen generation upon photo-excitation. Based on similar mechanisms to organic photosensitizers, organic sonosensitizers with long-lived triplet states are also favorable for SDT under US irradiation. Interestingly, phosphorescent p–n-CDs have long-lived triplet states in aqueous solutions, enabling effective singlet oxygen generation for SDT like organic photosensitizers. Unlike organic sonosensitizers with low photostability, p–n-CDs have excellent photostability owing to the graphitic structure. Moreover, p–n-CDs possess a narrow bandgap for enhanced sonoexcitation.

In conclusion, we have developed a one-step microwave strategy for controlled synthesis of NIR-phosphorescent p–n-CDs finely decorated with sulfonic acceptors and pyrrolic/graphitic N donors. The p–n-CDs are endowed with a narrow bandgap of 1.62 eV and a long carrier lifetime of 11.4 μs, two of which can enhance SDT as thermodynamically and dynamically favorable factors under low-intensity ultrasound irradiation, respectively. Additionally, p–n-CDs can emit NIR emission with PL maximum at 760 nm and maximum excitation at 700 nm with absolute PL quantum yield of 17.6% and improved stability against photo-bleaching. They exhibit enhanced $^1O_2$ generation efficiency owing to (1) inhibition of the e$^-$–h$^+$ pair recombination owing to the p–n junction, (2) long-lived triplet-state mediated $^1O_2$ generation, (3) GSH depletion using overexpressed GSH as a hole sacrificial agent. We have also prepared two kinds of p–n-CD loaded CCM vehicles with homotypic recognition capability as the cancer-specific NIR probe and sonosensitizer for precision bioimaging and SDT with excellent tumor eradication capability. Our results will open up a promising approach to engineer phosphorescent materials with NIR emission and long-lived triplet excited states for wide range applications apart from sonodynamic precision tumor therapy.

## Methods

### Materials
ICG and BPEI (Mn ∼ 1800 Da) were purchased from J&K Scientific Co., Ltd. (Beijing, China). Pyrene was obtained from TCI (Japan). Nitric acid ($HNO_3$) and PVA were purchased from Sinopharm Chemical Reagents Co., Ltd. (Shanghai, China). DPBF, 3-(4,5-dimethyl-thiazole-2-yl)−2,5-phenyltetrazolium bromide (MTT), dihydroethidium (DHE), and TEMP were purchased from Sigma-Aldrich.

### Preparation of p–n-CDs, p-CDs, and n-CDs
For the preparation of p–n-CDs, ICG (5 mg) and BPEI (50 mg) were dissolved in ethanol (10 mL) and then transferred to a reaction tube. The microwave reaction was performed at 200 °C for 15 min to obtain p–n-CDs, which were then dialyzed for 48 h to remove remaining ICG and BPEI. Similarly, p-CDs were synthesized by the microwave reaction at 200 °C for 15 min using single ICG as the precursor. For n-CDs, 1,3,6-trinitropyrene (TNP) was initially synthesized through the nitration reaction of pyrene at 80 °C for 48 h. The obtained TNP was filtered through a 220 μm microporous membrane for several times to remove the unreacted pyrene and $HNO_3$. Then, n-CDs were prepared through the microwave reaction at 200 °C for 15 min using TNP and BPEI as the precursors.

### Preparation of p–n-CD@PVA composite films
0.2 g of PVA was first added to 2 mL of DI water. To obtain uniform p–n-CD@PVA solution, p–n-CD aqueous solution (5 mg/mL, 100 μL) was

added to PVA solution (100 mg/mL) through a slow shake. The composite was coated onto the glass substrates and dried at 60 °C for 1 h to obtain the p−n-CD@PVA composite films.

## Preparation of CCM fragments

143B and Hela cells were purchased from Cell Bank of Type Culture Collection of Chinese Academy of Sciences (Shanghai, China). 143B and Hela cells were harvested and then resuspended in cold PBS. The collected cells were suspended in a hypotonic lysing buffer containing PMSF (Beyotime, China) and membrane protein extraction reagent. The resuspended 143B or Hela cells were incubated in ice bath for 15 min and then broken repeatedly using a freeze-thaw method. After centrifugation at $800 \times g$ for 10 min, the supernatant was further centrifuged at $15,000 \times g$ for 30 min to obtain the cell membrane fragments. The membrane products of CCM-143B and CCM-Hela were lyophilized and stored at −80 °C. The lyophilized membranes are rehydrated in ultrapure water or PBS buffer. For CCM protein characterization, the following antibodies were used for western blot analysis: Anti-pan cadherin antibody (Abcam, ab22744, 1:1000); Anti-Na+/K+-ATPase antibody (Abcam, ab76020, 1:1000), Anti-gp100 antibody (Abcam, ab137078, 1:1000), Anti-Histone H3.3 Rabbit pAb (Servicebio, GB11026, 1:500), Anti-COX IV Rabbit pAb (Servicebio, GB11250, 1:500), Anti-GAPDH Mouse mAb (Servicebio, GB12002, 1:1000), HRP conjugated Goat Anti-Rabbit IgG (Servicebio, GB23303, 1:3000), HRP conjugated Goat Anti-Mouse IgG (Servicebio, GB23301, 1:3000).

## Preparation of p−n-CD@CCM

A successive extrusion approach was utilized to prepare the p−n-CD@CCM. Briefly, 1 mL aqueous solution of p−n-CDs with varied concentration (0.1, 0.2, 0.3, 0.4, or 0.5 mg/mL) was mixed with CCM-143B or CCM-Hela dispersions (1 mL, 1.0 mg/mL). The p−n-CDs and CCM were transferred into a syringe and successively extruded through 500 nm, 200 nm, and 100 nm water-phase filters, followed by centrifugation to remove unloaded p−n-CDs from the dispersions. The amount of encapsulated p−n-CDs in p−n-CD@CCM was determined by UV-vis-NIR absorption spectroscopy. Finally, a p−n-CD@CCM dispersion with encapsulated p−n-CDs at a high concentration (200 μg/mL) and a high loading (-20%) was obtained at the mass ratio of p−n-CD to CCM (3:10 w/w) for further structural characterization and SDT measurements.

## Characterization

TEM images of p-CDs, n-CDs, p−n-CDs, and p−n-CD@CCM were acquired by a JEM-2100F microscope. XPS, FT-IR, XRD, and Raman spectra of p-CDs, n-CDs, and p−n-CDs were performed using Kratos Axis Ultra DLD, Nicolet AVATAR 370, Rigaku 18 KW D/max-2550, and Renishaw in plus laser Raman spectrometer, respectively. The hydrodynamic diameter and Zeta potential measurements were carried out using a Malvern Zetasizer Nano ZS90. The UV-vis-NIR absorption and fluorescence spectra of p-CDs, n-CDs, and p−n-CDs were measured by Agilent Cary 5000 and Hitachi 7000 spectrophotometer. The phosphorescence spectra and lifetimes were measured by an Edinburgh FLS1000 spectrometer. ESR spectra were recorded on a Bruker EMX plus ESR spectrometer.

## Computational details

We first established three $sp^2$ hybrid carbon clusters with different functional groups as CD-like molecule models including SO$_3$-modified CD-1, SO$_3$-pyrrolic N-modified CD-2, and SO$_3$-pyrrolic N-graphitic N-modified CD-3. The geometry of CD molecules at the ground state was optimized with dispersion corrected density functional theory (DFT-D3)[67,68]. To investigate the photophysical properties of CD molecules, their excited electronic structures were calculated at the PBE0-D3/TZVP level with the TD-DFT method. Their UV absorption spectra were

obtained by broadening the oscillator strength at corresponding excited energy using Gauss broadening functions. According to Kasha's rule, their fluorescence spectra are related to the radiation transition of the first singlet excited state (S1). In order to simulate the emission spectra of CD molecules, the molecular geometries at the S1 excited state were then optimized at the PBE0/TZVP level with the TD-DFT method. All the calculations were performed using the Gaussian 16 program.

## Sonodynamic performance of p−n-CDs

The mixture solution of 20 μg DPBF and p−n-CD (200 μg/mL, 1 mL) were exposed to US irradiation (50 kHz, 3.0 W/cm$^2$) for 5 min and the absorbance of DPBF at 418 nm was recorded to calculate the quantum yield of $^1O_2$ generation. Similarly, the $^1O_2$ generation efficiency of p-CDs, n-CDs, ICG, and commercial TiO$_2$ nanopaticles were measured under the same conditions. For ESR measurements, the mixture solution of 20 μL TEMP and p−n-CD (200 μg/mL, 1 mL) were exposed to US irradiation for 2 min and the signal of $^1O_2$ was detected by ESR spectrometer using TEMP as the trapping agent. Similarly, the ESR spectra of US alone, p-CDs, n-CDs, ICG, and commercial TiO$_2$ nanopaticles were detected under the same conditions.

## Electrochemical measurements

The CV curves of p−n-CDs, p-CDs, and n-CDs were measured based on a three-electrode system using electrochemical workstation (CHI 660D, China). The reference electrode, counter electrode, and electrolyte were Ag/AgCl, platinum wire, and TBAPF$_6$, respectively. According to the CV curves, the conduction band (CB) energy level of p−n-CDs, p-CDs, and n-CDs at pH 7.4 was measured to be −0.77, −1.50, and −1.36 eV vs Ag/AgCl, respectively. The conductivity type of p−n-CDs, p-CDs, and n-CDs was analyzed through impedance spectroscopy equipped with Thales software. The conduction- and valence-band edges of p−n-CDs, p-CDs, and n-CDs were determined by a linear potential scan (5 mV/s).

## MTT assay

143B cells were first seeded in the 96-well plates and incubated for 24 h. After incubated with p−n-CD@CCM for another 24 or 48 h, a standard MTT assay was performed to detect the cell viability. For in vitro sonodynamic therapy, after incubated with p−n-CD@CCM for 4 h, 143B cells were exposed to US irradiation (50 kHz, 3.0 W/cm$^2$) for 5 min. Then, 143B cells were incubated for 24 or 48 h and a standard MTT assay was performed to detect the cell viability.

## Live/dead cell staining

Calcein AM and PI were used as the probes to stain the live and dead cells after treating with PBS, US, p−n-CD@CCM, ICG + US, p−n-CD + US, or p−n-CD@CCM + US. US treatments were carried out for 5 min under the condition of 50 kHz and 3.0 W/cm$^2$. The fluorescence microscope (Olympus, Japan) was utilized to obtain the fluorescence images under the excitation wavelength of 488 nm for calcein AM and 561 nm for PI. The semi-quantitative analysis of live/dead cell staining images was determined by the Image J software (ver. 1.7) via triplicate parallel samples.

## In vitro ROS detection

For in vitro ROS detection, DCFH-DA was used to detect intracellular ROS level in the 143B cells after treating with PBS, US, p−n-CD@CCM, ICG + US, p−n-CD + US, or p−n-CD@CCM + US. US treatments were carried out for 5 min under the condition of 50 kHz and 3.0 W/cm$^2$. The confocal images were collected using Olympus FV3000 microscope and the data were analyzed with FV31S-SW (ver. 2.1.1.98). The semi-quantitative analysis of ROS staining images was determined by the Image J software (ver. 1.7) via triplicate parallel samples.

## Apoptosis assay

For cell apoptosis analysis, the 143B cells were first seeded in 6-well plates and then incubated overnight. After different treatments including control, US alone, p–n-CD@CCM, ICG + US, p–n-CD + US, and p–n-CD@CCM + US, the 143B cells were incubated for another 24 h. US treatments were carried out for 5 min under the condition of 50 kHz and 3.0 W/cm². Then, 143B cells were collected by centrifugation ($300 \times g$, 5 min) and detected by flow cytometer (Beckman, CytoFLEX LX) using an Annexin V-FITC/PI assay kit (Dojindo, Japan) according to the manufacturer's protocol. For the detailed operation, 143B cells were first stained with Annexin V-FITC (5 μL) for 15 min in Annexin-binding buffer. PI (5 μL) was then added to 143B cells and incubated for another 15 min. Beckman Coulter cytExpert 2.3.0.84 software was used for data collection and analysis.

## Cell imaging

For in vitro cell imaging, 143B and Hela cells were first seeded in 96-well plates. The p–n-CD@143B and p–n-CD@Hela solutions were introduced into the 143B or Hela cells with a final concentration of ~10 μg mL⁻¹, respectively. After 30 min of incubation, the 143B and Hela cells were washed with PBS for three times to remove excessive p–n-CD@143B or p–n-CD@Hela. Afterward, the cells were fixed with 4% paraformaldehyde solution and stained by 4',6-diamidino-2-phenylindole (DAPI), followed by observation under a confocal microscope (Olympus, Japan). Confocal imaging data were analyzed with FV31S-SW (ver. 2.1.1.98).

## Tumor model

3–5-week-old female Balb/c nude mice were purchased from SLAC Laboratory Animal (Shanghai, China). All mice were housed in an animal facility under constant environmental conditions (room temperature, $22 \pm 1\,°C$, relative humidity, 40–70% and a 12 h light-dark cycle). All animal experiments were carried out under the permission by Institutional Animal Care and Use Committee of Shanghai University (SYXK 2019-0020). To establish a tumor model on the nude mice, $5 \times 10^6$ of 143B cells were subcutaneously injected into their back side.

## In vivo NIR fluorescence imaging

The p–n-CD@CCM (2.0 mg kg⁻¹) was intravenously injected to 143B tumor-bearing mice. After intravenous injection of different time, an IVIS Lumina III in vivo Imaging System (PerkinElmer, USA) was utilized to obtain the NIR fluorescence images under the excitation and collection wavelength of 700 and 790 nm, respectively. The NIR fluorescence imaging has been repeated for three times and the data were analyzed with Living Image 4.5.2 Software (PerkinElmer, USA).

## In vivo SDT

Six groups of 143B tumor-bearing mice were randomly divided as followed: control, US, p–n-CD@CCM, ICG + US, p–n-CD + US, and p–n-CD@CCM + US. After intravenous injection of 24 h, US irradiation was conducted in groups 2, 4, 5, and 6 for 5 min under the condition of 50 kHz and 3.0 W/cm². We measured the tumor sizes and body weight of mice every two days and every day, respectively. For in vivo deep-tissue SDT, we initially established two 143B tumors on either side of mice and the US irradiation was conducted from left to right. On the 2-day post-treatment, tumors in each groups were harvested for histological analysis including H&E and ROS staining. Tumors in each groups were also collected for ROS detection using DHE as the probe. On the 16-day post-treatment, major organs and blood in each groups were harvested to assess the in vivo long-term toxicity through histological analysis, blood biochemistry, and blood routine analysis.

## In vivo pharmacokinetic

3–5-week-old female Balb/c mice ($n = 5$) were intravenously injected with p–n-CDs and p–n-CD@CCM, respectively, at a p–n-CD dose of 2.0 mg/kg. 10 μL of blood was collected in heparinized tubes from the tail vein at different time points including 5 min, 10 min, 30 min, 1 h, 2 h, 4 h, 8 h, 12 h, and 24 h, dispersed into 990 μL PBS, and centrifuged at $1500 \times g$ for 10 min to harvest plasma samples. The p–n-CD concentration in the supernatants was examined by chromatometry at 680 nm. The plasma concentration-time profiles of p–n-CD and p–n-CD@CCM samples showed a typical double-compartment pharmacokinetic behavior.

## In vivo metabolism study

After intravenous injection of p–n-CD@CCM (2.0 mg/kg), 3–5-week-old female Balb/c mice ($n = 5$) were sacrificed at 1, 7, and 14 days, respectively. The major organs were collected and solubilized by aqua regia for detection of p–n-CD concentrations by chromatometry at 680 nm. To study the excretion pathway, after intravenous injection of p–n-CD@CCM (2.0 mg/kg), Balb/c mice ($n = 5$) were kept in metabolic cages to collect their feces and urine. The feces and urine were solubilized by aqua regia and the 680 nm absorbance was measured to determine the concentration of p–n-CD.

## Statistical analysis

To ensure the accuracy of the experiments, at least three replicates were performed. All data are presented as mean values ± SD. All statistical analyses were performed by Origin 2018 and Graphpad Prism 8.0. Statistical significance between two groups was calculated with two-tailed Student's $t$ test.

## Reporting summary

Further information on research design is available in the Nature Research Reporting Summary linked to this article.

## Data availability

The authors declare that all relevant data supporting the findings of this study are available within the paper, the Supplementary Information, and Source data. Source data are provided with this paper.

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

## Acknowledgements

This work was supported by the National Natural Science Foundation of China (No. 22278262 to D.P., 22108168 to B.G., 12175142 to L.S., 22177067 to L.F.), Shanghai Rising-Star Program (A type) (No. 20QA1403400 to L.F.), China Postdoctoral Science Foundation (No. 2020M681267 to B.G.), the Science and Technology Commission of Shanghai Municipality (No. 22ZR1424000 to D.P.), and the cultivation project (No. ynms202103 to L.S.) from Shanghai Jiao Tong University affiliated Sixth People's Hospital.

## Author contributions

D.P. designed and directed the study, analyzed the data and wrote the manuscript. B.G. designed and performed experiments, and collected and analyzed data. L.F., L.S. provided important suggestions, supervised parts of the project, and improved the manuscript. S.F. provided advice and technical help with cell experiments and mouse cancer models. J.H., Y.L. performed cell and animal experiments. All authors discussed the results and reviewed the manuscript.

## Competing interests

The authors declare no competing interests.
