## [Peer Review File · Nature Communications]

Near-infrared phosphorescent carbon dots for sonodynamic precision tumor therapyReviewers' comments:

Reviewer #1 (Remarks to the Author):

The manuscript explored a NIR-emitting sonosensitizers from a NIR phosphorescent carbon dot (CD) material to enhance SDT as thermodynamically and dynamically favorable factors under low-intensity ultrasound irradiation. Although the manuscript demonstrated the medical application of NIR-phosphorescent materials as long-lived probes and enhanced sonosensitizers, the subject design was not pursued from a pharmaceuticals perspective. In addition, the data in the manuscript were not reasonably analyzed. Therefore, I think the manuscript cannot be accepted by Nature Communications. Below are some comments for the authors' consideration.

Major points:

1. p-n-CDs showed superior NIR emitting stability in aqueous solutions stored for 7 days. After CMM coating, how is the stability of p-n-CD@CCM? Would the size and Zeta potential change in aqueous solutions? And further in physiological environment such as phosphate buffered solution and serum?
2. The manuscript lacks in vivo pharmacokinetic examination of p-n-CD@CCM.
3. The manuscript lacks in vivo metabolic process study of p-n-CD@CCM.
4. The TEM images of p-n-CD@CCM in Fig. 5a was larger than 100 nm, but the DLS size distribution of p-n-CD@CCM were 78.8 ± 1.3 nm. I doubt the authenticity of the results. Please show a broader view of TEM results of p-n-CD@CCM to demonstrate the universality. Moreover, the size of p-n-CDs was ~ 2 nm, but the hydrodynamic diameter of p-n-CD@CCM vehicles was 78.8 ± 1.3 nm. How many p-n-CDs were encapsulated in p-n-CD@CCM? Which aspect was considered to select the particle size of 78.8 ± 1.3 nm? The authors should explain it. Would the the particle size affect the in vivo behavior of materials?
5. From Supplementary Fig. 19, the mice in the final formulation group were already tumor free, where is the tissue from for the ROS staining and H&E staining experiments in Fig. 6i and 6j?
6. In Supplementary Fig. 16, the NIR fluorescence signal intensity lacks error bars. Also, the NIR fluorescence signal intensity lacks error bars in Supplementary Fig. 17b. How many times was the experiment repeated?

Minor points:

1. Figure legends of Fig. 5k and 5l were mistaken.
2. Fig. 5e had no fluorescence images of tumor cells such as DAPI, how was it determined whether the control group had tumor cells?
3. Fig. 5k, 5l and Supplementary Fig. 14 lacks quantitative analysis.
4. Fig. 6d and Supplementary Fig. 18 lacks fluorescence quantification of tissues.
5. *** was marked in Figure 6e and 6f. But the authors did not indicate that which two groups were compared.
6. Supplementary Fig. 15 lacks quantitative or semi-quantitative analysis of fluorescence intensity in tumor cells.
7. The authors should carefully check the format of references. Reference 24 lacks volume number. Moreover, many references had capitalized titles, such as Reference 19, 22, and 26.

Reviewer #2 (Remarks to the Author):

In this manuscript "Near-infrared phosphorescent carbon dots for sonodynamic precision tumor therapy", the authors developed a metal-free carbon dot (CDs) as a theranostic platform for NIR imaging guided sonodynamic therapy. The results showed significant regression of tumor in vivo model based on combinational effects of cancer-specific targeting and sonodynamic therapy (SDT) with complete eradication of solid tumors. The results are interesting. However, more intensive discussions and experiments are needed.

1. ICG is a typical tricyanocyanine dye. However, the p-type CDs (p-CDs) prepared by microwave treatment of ICG showed no fluorescence, why? The NIR phosphorescence mechanisms of p-n-CDs should be discussed.
2. The sonodynamic mechanism of p-n-CDs needs more careful investigation and explanation. It is not clear why p-n-CDs can generate ROS under US.
3. What is the stability and blood circulation behavior of p-n-CDs without any modification in physiological conditions? Please add the related data.
4. The antitumor mechanism of p-n-CDs in cellular level should be explored.
5. The results in Figure 6b and 6c are the in vivo NIR imaging of mice, which shows that there was obvious fluorescence signal in the liver at 0 h post-injection, why?
6. In the in vivo phase, the data shows only one round of treatment is enough to inhibit tumor growth. Whether tumor cells in tissue were apoptosis on day 4 or 5, or other long-term antitumor mechanism? Please explain it with data.
7. The legend in Figure 6g and 6h should be checked again.

Itemized Responses to Reviewers' Comments

Reviewer #1

The manuscript explored a NIR-emitting sonosensitizers from a NIR phosphorescent carbon dot (CD) material to enhance SDT as thermodynamically and dynamically favorable factors under low-intensity ultrasound irradiation. Although the manuscript demonstrated the medical application of NIR-phosphorescent materials as long-lived probes and enhanced sonosensitizers, the subject design was not pursued from a pharmaceuticals perspective. In addition, the data in the manuscript were not reasonably analyzed. Therefore, I think the manuscript cannot be accepted by Nature Communications. Below are some comments for the authors' consideration.

Reply: Thanks for the reviewer's careful, valuable suggestions from pharmaceuticals perspectives. Indeed, some important pharmaceuticals data needed be provided in terms of physiological stability and pharmacokinetic properties. In the revised version, we added these pharmaceuticals data and addressed the reviewer's all comment points as followed.

Major points:

- 1. p-n-CDs showed superior NIR emitting stability in aqueous solutions stored for 7 days. After CMM coating, how is the stability of p-n-CD@CCM? Would the size and Zeta potential change in aqueous solutions? And further in physiological environment such as phosphate buffered solution and serum?**

Reply: The high physiological stability for a newly designed drug is of importance in clinical applications. The stability of p-n-CD@CCM as a new class of metal-free sonosensitizers was thus examined in PBS and fetal bovine serum (FBS). As shown in Supplementary Fig. 16a, no visible precipitate was observed in PBS and FBS solutions after storing for 7 days, demonstrating the outstanding stability of p-n-CD@CCM in the physiological environment for medical applications. The high stability of p-n-CD@CCM in aqueous solution was further evaluated by measurement of change in hydrodynamic diameter and Zeta potential during storage. There was no substantial change in hydrodynamic diameter and Zeta potential after storing for 7 days (Supplementary Fig. 16b, c).

We have added these results and corresponding discussion in the revised manuscript.

Supplementary Fig. 16 (a) Photographs of p-n-CD@CCM aqueous solution, PBS solution, and fetal bovine serum (FBS) solution stored for different periods of time (0, 1, and 7 days). (b, c) Hydrodynamic diameters (b) and Zeta potential (c) of p-n-CD@CCM aqueous solution stored for different periods of time (0, 1, and 7 days).

2. The manuscript lacks *in vivo* pharmacokinetic examination of p-n-CD@CCM.

Reply: In the previous version, we explored the *in vivo* pharmacokinetics properties of p-n-CD@CCM only in terms of *in vivo* drug distribution via *in vivo* and Ex vivo NIR imaging, which revealed that the CCM coating enabled the high-level intratumor enrichment of the theranostic p-n-CDs as compared with free p-n-CDs without coating (Fig. 6b, c).

In the current version, the *in vivo* pharmacokinetics of p-n-CD@CCM was further explored by measuring plasma concentration-time profiles. Female Balb/c mice (n=5) were i.v. injected with p-n-CDs and p-n-CD@CCM, respectively, at a p-n-CD dose of 2.0 mg/kg. 10 μ L of blood was collected in heparinized tubes from the tail vein at 5 min, 10 min, 30 min, 1 h, 2 h, 4 h, 8 h, 12 h, and 24 h post injection, dispersed into 990 μ L PBS, and centrifuged at 5000 rpm for 10 min to harvest plasma samples. The p-n-CD concentration in the supernatants was examined by chromatometry at 680 nm. The plasma concentration-time profiles of p-n-CD and p-n-CD@CCM samples showed a typical double-compartment pharmacokinetic behavior (Supplementary Fig. 26). For free p-n-CDs without CCM coating, the elimination half-life ($t_{1/2}$) was only 0.8 ± 0.1 h, revealing rapid renal clearance owing

to the ultrafine size (2.2 ± 0.4 nm), like previous reports on CDs. In contrast, after CCM coating, the half-life was elongated to 11.6 ± 0.9 h, 14.5 times of that of free p-n-CDs, indicating that the typical renal clearing process of CDs was inhibited by CCM coating. The longer circulation time allowed p-n-CD@CCM to enrich in tumors at high levels.

We have added these results and corresponding discussion in the revised manuscript.

Supplementary Fig. 26 Plasma concentration-time profiles (a) and pharmacokinetic parameters (b) of p-n-CD and p-n-CD@CCM after intravenous administration of various drug formulations at the p-n-CD dose of 2.0 mg/kg (n=5).

3. The manuscript lacks *in vivo* metabolic process study of p-n-CD@CCM.

Reply: The *in vivo* biodistribution and excretion pathway of p-n-CD@CCM were explored for examination of its *in vivo* metabolic process. Female Balb/c mice (n=5) after i.v. injection with p-n-CD@CCM (2.0 mg/kg) was sacrificed at 1, 7, and 14 days, respectively. The major organs were collected for detection of p-n-CD concentrations by chromatometry at 680 nm after these organs were solubilized by aqua regia. Supplementary Fig. 27a revealed that p-n-CD@CCM primarily accumulated in liver at 24 h post-injection due to the capture of the reticuloendothelial system. Importantly, no obvious p-n-CD signals were observed in these organs after 14 days, suggesting the nearly complete clearance of p-n-CD@CCM. To study the excretion pathway, mice after i.v. injection with p-n-CD@CCM (2.0 mg/kg) were kept in metabolic cages to collect their feces and urine at various time points, which were solubilized by aqua regia measured by chromatometry at 680 nm to determine p-n-CD concentration. Much higher levels of p-n-CD were observed in the feces, demonstrating that p-n-CD@CCM was eliminated from the mice majorly by liver excretion (Supplementary Fig. 27b).

We have added these results and corresponding discussion in the revised manuscript.

Supplementary Fig. 27 (a) Biodistribution of p-n-CD@CCM post i.v. injection in mice on different days (n=5). (b) The detected p-n-CD mass in urine and feces at different time points post i.v. injection of p-n-CD@CCM (n=5).

4. The TEM images of p-n-CD@CCM in Fig. 5a was larger than 100 nm, but the DLS size distribution of p-n-CD@CCM were 78.8 ± 1.3 nm. I doubt the authenticity of the results. Please show a broader view of TEM results of p-n-CD@CCM to demonstrate the universality. Moreover, the size of p-n-CDs was ~ 2 nm, but the hydrodynamic diameter of p-n-CD@CCM vehicles was 78.8 ± 1.3 nm. How many p-n-CDs were encapsulated in p-n-CD@CCM? Which aspect was considered to select the particle size of 78.8 ± 1.3 nm? The authors should explain it. Would the particle size affect the in vivo behavior of materials?

Reply: (1) According to the reviewer's suggestion, we provided a broad-view TEM image of p-n-CD@CCM. As shown in Supplementary Fig. 14, for most p-n-CD@CCM particles, their sizes were less than 100 nm, which was consistent with the DLS result.

Supplementary Fig. 14 TEM image of p-n-CD@CCM.

(2) How many p-n-CDs were encapsulated in p-n-CD@CCM? It is technically difficult to determine the number of p-n-CDs encapsulated in p-n-CD@CCM because we cannot observe and count all p-n-

CDs encapsulated in a single p-n-CD@CCM particle. Yet, the loading rate in p-n-CD@CCM can be determined to be 20%.

(3) Previous reports have demonstrated that suitable nanoparticle size in the range of 10-100 nm is a key factor that is advantageous to tumor tissue accumulation and tumor penetration (*Adv. Mater.*, 2016, 28, 7340-7364; *J. Am. Chem. Soc.*, 2020, 142, 6527-6537; *ACS Nano*, 2020, 14, 15119-15130). Therefore, we believed that the suitable size of p-n-CD@CCM was controlled to be 78.8 ± 1.3 nm through a 100 nm water-phase filter by an extrusion approach, which is of importance in high-level intratumoral accumulation and deep intratumoral infiltration.

We have added these results and corresponding discussion in the revised manuscript.

5. From Supplementary Fig. 19, the mice in the final formulation group were already tumor free, where is the tissue from for the ROS staining and H&E staining experiments in Fig. 6i and 6j?

Reply: For the best treatment group p-n-CD@CCM+ US, the tumor growth was inhibited gradually, and finally the tumor was eradicated after 12 days. So we selected a time point at the second day post treatment for H&E and ROS staining.

6. In Supplementary Fig. 16, the NIR fluorescence signal intensity lacks error bars. Also, the NIR fluorescence signal intensity lacks error bars in Supplementary Fig. 17b. How many times was the experiment repeated?

Reply: According to the reviewer's suggestion, we have repeated the *in vivo* NIR fluorescence imaging for three times and the error bars have been added in the Supplementary Fig. 20, 21b.

Supplementary Fig. 20 NIR fluorescence signal intensity of the left tumor (Hela) and right tumor

(143B) at 24 h post intravenous injection with p-n-CDs (a) or p-n-CD@143B (b).

Supplementary Fig. 21 (a) *In vivo* NIR fluorescence images at 24 h post intravenous injection with p-n-CD@HeLa. (b) NIR fluorescence signal intensity of the left tumor (HeLa) and right tumor (143B) at 24 h post intravenous injection with p-n-CD@HeLa.

Minor points:

1. Figure legends of Fig. 5k and 5l were mistaken.

Reply: Thanks for your careful work. We have modified the Figure legends of Fig. 5k and 5l in the revised manuscript.

2. Fig. 5e had no fluorescence images of tumor cells such as DAPI, how was it determined whether the control group had tumor cells?

Reply: According to the reviewer's suggestion, we performed the confocal imaging using DAPI to stain the cell nucleus and the p-n-CD@CCM probe to stain the cytoplasm, as shown in Fig. 5e, f.

Fig. 5 (e, f) Cancer-specific confocal images of 143B and HeLa cells incubated with p-n-CD@HeLa or

p-n-CD@143B, respectively.

3. Fig. 5k, 5l and Supplementary Fig. 14 lacks quantitative analysis.

Reply: According to the reviewer's suggestion, the quantitative analysis of live/dead cell staining and ROS staining images was determined by the Image J software via triplicate parallel samples. We have added the quantitative results in the revised Supplementary Fig. 19.

Supplementary Fig. 19 The quantitative results of live/dead cell staining (a) and ROS staining (b) images determined by the Image J software via triplicate parallel samples

4. Fig. 6d and Supplementary Fig. 18 lacks fluorescence quantification of tissues.

Reply: According to the reviewer's suggestion, the quantitative analysis of NIR fluorescence intensity of p-n-CD@143B and p-n-CD@Hela in the major organs and tumor tissues have been added in the revised Supplementary Fig. 23.

Supplementary Fig. 23 Time-dependent NIR fluorescence intensity of p-n-CD@143B (a) and p-n-CD@Hela (b) in the major organs and tumor tissues based on the fluorescence imaging results.

5. * was marked in Figure 6e and 6f. But the authors did not indicate that which two groups**

were compared.

Reply: According to the reviewer's suggestion, the statistical differences between the control group and the p-n-CD@CCM + US group have been marked in Fig. 6e and 6f.

Fig. 6 (e) 143B tumor growth curves of mice after indicated treatments. (f) 143B tumor weight after indicated treatments.

6. Supplementary Fig. 15 lacks quantitative or semi-quantitative analysis of fluorescence intensity in tumor cells.

Reply: According to the reviewer's suggestion, the quantitative analysis of ROS staining images was determined by the Image J software via triplicate parallel samples. We have added the quantitative results in the revised Supplementary Fig. 19b.

Supplementary Fig. 19 (b) The quantitative results of ROS staining images determined by the Image J software via triplicate parallel samples

7. The authors should carefully check the format of references. Reference 24 lacks volume number. Moreover, many references had capitalized titles, such as Reference 19, 22, and 26.

Reply: Thanks for your careful work. We have modified the format of references in the revised

manuscript.

Reviewer #2

In this manuscript “Near-infrared phosphorescent carbon dots for sonodynamic precision tumor therapy” , the authors developed a metal-free carbon dot (CDs) as a theranostic platform for NIR imaging guided sonodynamic therapy. The results showed significant regression of tumor in vivo model based on combinational effects of cancer-specific targeting and sonodynamic therapy (SDT) with complete eradication of solid tumors. The results are interesting. However, more intensive discussions and experiments are needed.

Reply: The detailed discussion on the NIR phosphorescent and sonodynamic mechanisms has been added in the Discussion section in the new version, as followed.

Recently, CDs have been engineered as promising fluorescent materials owing to their numerous merits such as high optical stability, excellent solubility, low toxicity, and easy preparation. Generally, single-component CDs do not exhibit observable phosphorescence due to their unstable triplet states. To extend their phosphorescent applications in data encryption and anti-counterfeiting, great effort has been made in the construction of CD-based hybrid phosphorescent systems, in which various organic or inorganic matrices, including $KAl(SO_4)_2 \cdot xH_2O$, SiO_2 , zeolites, layered double hydroxides, poly(vinyl alcohol), polyurethane, and cyanuric acid (*J. Mater. Chem. C*, 2015, 3, 2798-2801; *Nat. Commun.*, 2020, 11, 559; *Sci. Adv.*, 2017, 3, e1603171; *Nanoscale*, 2017, 9, 6658; *Angew. Chem. Int. Ed.*, 2016, 55, 7231-7235; *Nanoscale*, 2016, 8, 4742; *J. Mater. Chem. C*, 2018, 6, 7890-7895), were employed to suppress the intermolecular vibration for triplet stabilization. However, CD-based hybrid phosphorescent systems are limited by the solid-state-only applications as well as time-consuming and high-cost fabrication involving the use of additional matrices in the solid composites. To date, matrix-free phosphorescent CDs have not been fabricated for phosphorescent applications in aqueous solutions or physiological media because several issues are difficult to address by self-stabilization of triplet states. First, a challenge is to substantially reduce the energy gap (ΔE_{ST}) of CDs between the lowest singlet (S1) to the lowest triplet (T1) for enhancing their favorable intersystem crossing. Second, highly efficient fluorescent emission in CD solutions shows overwhelming advantages over phosphorescent emission, leading to extremely low population probability in the triplet states despite

effective ISC. Unfortunately, the electron–hole recombination from S1 to the ground state could not be inhibited to quench the fluorescence emission for enhanced RTP. Third, the excited triplet states at a low-level population are strongly quenched by dissolved molecular oxygen (triplet-state $^3\text{O}_2$) to form singlet-state $^1\text{O}_2$, further impeding the aqueous applications of RTP materials. Herein, we reported for the first time that matrix-free p-n-CDs in solution states can be rationally constructed as a solution-applicable RTP material by p-n junction engineering. The formation of the p-n junction within single p-n-CDs can effectively inhibit the e^-h^+ pair recombination from excited singlet states to the ground state and thus enhance the RTP emission as evidenced by the time-resolved PL spectra of the p-n-CD solution and the p-n-CD@PVA film, where no transient fluorescent components were observed apart from the long-lived components. In addition, rapid charge transfers from the n-type dopant (pyrrolic N) to the p-type dopant (SO_3^-) in the bipolar configuration could stabilize the triplet states with a long lifetime of 11.4 μs in aqueous solutions. In contrast, for p-CDs with SO_3^- functionalization only or n-CDs with pyrrolic/graphitic N functionalization only, effective charge transfer could not be established, leading to fluorescent emission with a ns-scale lifetime in aqueous solutions. Furthermore, TD-DFT calculations showed that the simultaneous modification with SO_3 , pyrrolic N, and graphitic N in the CD-3 model can reduce ΔE_{ST} to a smaller value (0.55 eV), which is favorable for effective ISC to populate triplet excitons and consequently emitting phosphorescence. In contrast, for SO_3 -modified CD-1 has ΔE_{ST} as high as 1.08 eV, suggesting ineffective ISC at room temperature. These results clearly revealed that phosphorescent p-n-CDs exhibited a distinct triplet stabilization mechanism by creating the p-n junction rather than the hydrogen-bonded network in CD-based hybrid phosphorescent systems. Compared with the complex preparation processes of CD-based hybrid phosphorescent systems, the construction of highly phosphorescent p-n junction CDs is simple, one-step, and rapid by one-pot microwave irradiation.

It is noted that current afterglow CDs in the hybrid systems are mainly limited to visible RTP emissions from blue, green to yellow. Recently, high-efficiency red RTP emissions at 620 nm have been also achieved by embedding CDs in Mn-containing open-frame-work matrices (Angew. Chem. Int. Ed., 2019, 58, 18443-18448), but RTP emissions extended to the NIR region remain unexplored to date for NIR applications such as NIR imaging and emitters. Generally, the NIR emissions of

quantum dots require a narrow bandgap; it is challenging for CDs to simultaneously reduce the bandgap and stabilize triplet states. Here, we found that phosphorescent p-n-CDs have a narrow bandgap of 1.62 eV and thus emit NIR phosphorescence at 760 nm as a result of the synergistic effects of electron-withdrawing and electron-donating modifications on the p-n-CD electronic structure. The rational p-n junction modification in NIR-phosphorescent p-n-CDs creates the narrow bandgap and stabilize triplet states. Importantly, the p-n-CDs exhibit a phosphorescent efficiency as high as 17.6% in aqueous solutions and high NIR emission brightness comparable to that of the clinically used ICG. These RTP properties render p-n-CDs as a superior long-lived NIR imaging platform for biomedical applications with minimal background interference.

Long-lived triplet states of organic photosensitizers are tremendously important for photodynamic therapy (PDT) owing to a high yield of singlet oxygen generation upon photo-excitation. Based on similar mechanisms to organic photosensitizers, organic sonosensitizers with long-lived triplet states are also favorable for SDT under US irradiation. Interestingly, phosphorescent p-n-CDs have long-lived triplet states in aqueous solutions, enabling effective singlet oxygen generation for SDT like organic photosensitizers. Unlike organic sonosensitizers with low photostability, p-n-CDs have excellent photostability owing to the graphitic structure. Moreover, p-n-CDs possess a narrow bandgap for enhanced sonoexcitation.

1. ICG is a typical tricyanocyanine dye. However, the p-type CDs (p-CDs) prepared by microwave treatment of ICG showed no fluorescence, why? The NIR phosphorescence mechanisms of p-n-CDs should be discussed.

Reply: As shown in Fig. 4a, p-CDs also emitted strong fluorescence rather than no fluorescence. Their fluorescence peak was at 430 nm rather than at 810 nm for ICG, which was correlated with the wider bandgap of p-CDs.

The NIR phosphorescence mechanisms of p-n-CDs should be discussed as followed: Recently, CDs have been engineered as promising fluorescent materials owing to their numerous merits such as high optical stability, excellent solubility, low toxicity, and easy preparation. Generally, single-component CDs do not exhibit observable phosphorescence due to their unstable triplet states. To extend their phosphorescent applications in data encryption and anti-counterfeiting, great effort has been made in

the construction of CD-based hybrid phosphorescent systems, in which various organic or inorganic matrices, including $\text{KAl}(\text{SO}_4)_2 \cdot x\text{H}_2\text{O}$, SiO_2 , zeolites, layered double hydroxides, poly(vinyl alcohol), polyurethane, and cyanuric acid (*J. Mater. Chem. C*, 2015, 3, 2798-2801; *Nat. Commun.*, 2020, 11, 559; *Sci. Adv.*, 2017, 3, e1603171; *Nanoscale*, 2017, 9, 6658; *Angew. Chem. Int. Ed.*, 2016, 55, 7231-7235; *Nanoscale*, 2016, 8, 4742; *J. Mater. Chem. C*, 2018, 6, 7890-7895), were employed to suppress the intermolecular vibration for triplet stabilization. However, CD-based hybrid phosphorescent systems are limited by the solid-state-only applications as well as time-consuming and high-cost fabrication involving the use of additional matrices in the solid composites. To date, matrix-free phosphorescent CDs have not been fabricated for phosphorescent applications in aqueous solutions or physiological media because several issues are difficult to address by self-stabilization of triplet states. First, a challenge is to substantially reduce the energy gap (ΔE_{ST}) of CDs between the lowest singlet (S1) to the lowest triplet (T1) for enhancing their favorable intersystem crossing. Second, highly efficient fluorescent emission in CD solutions shows overwhelming advantages over phosphorescent emission, leading to extremely low population probability in the triplet states despite effective ISC. Unfortunately, the electron-hole recombination from S1 to the ground state could not be inhibited to quench the fluorescence emission for enhanced RTP. Third, the excited triplet states at a low-level population are strongly quenched by dissolved molecular oxygen (triplet-state $^3\text{O}_2$) to form singlet-state $^1\text{O}_2$, further impeding the aqueous applications of RTP materials. Herein, we reported for the first time that matrix-free p-n-CDs in solution states can be rationally constructed as a solution-applicable RTP material by p-n junction engineering. The formation of the p-n junction within single p-n-CDs can effectively inhibit the e^-h^+ pair recombination from excited singlet states to the ground state and thus enhance the RTP emission as evidenced by the time-resolved PL spectra of the p-n-CD solution and the p-n-CD@PVA film, where no transient fluorescent components were observed apart from the long-lived components. In addition, rapid charge transfers from the n-type dopant (pyrrolic N) to the p-type dopant (SO_3^-) in the bipolar configuration could stabilize the triplet states with a long lifetime of 11.4 μs in aqueous solutions. In contrast, for p-CDs with SO_3^- functionalization only or n-CDs with pyrrolic/graphitic N functionalization only, effective charge transfer could not be established, leading to fluorescent emission with a ns-scale lifetime in aqueous solutions. Furthermore, TD-DFT

calculations showed that the simultaneous modification with SO₃, pyrrolic N, and graphitic N in the CD-3 model can reduce ΔE_{ST} to a smaller value (0.55 eV), which is favorable for effective ISC to populate triplet excitons and consequently emitting phosphorescence. In contrast, for SO₃-modified CD-1 has ΔE_{ST} as high as 1.08 eV, suggesting ineffective ISC at room temperature. These results clearly revealed that phosphorescent p-n-CDs exhibited a distinct triplet stabilization mechanism by creating the p-n junction rather than the hydrogen-bonded network in CD-based hybrid phosphorescent systems. Compared with the complex preparation processes of CD-based hybrid phosphorescent systems, the construction of highly phosphorescent p-n junction CDs is simple, one-step, and rapid by one-pot microwave irradiation.

It is noted that current afterglow CDs in the hybrid systems are mainly limited to visible RTP emissions from blue, green to yellow. Recently, high-efficiency red RTP emissions at 620 nm have been also achieved by embedding CDs in Mn-containing open-frame-work matrices (Angew. Chem. Int. Ed., 2019, 58, 18443-18448), but RTP emissions extended to the NIR region remain unexplored to date for NIR applications such as NIR imaging and emitters. Generally, the NIR emissions of quantum dots require a narrow bandgap; it is challenging for CDs to simultaneously reduce the bandgap and stabilize triplet states. Here, we found that phosphorescent p-n-CDs have a narrow bandgap of 1.62 eV and thus emit NIR phosphorescence at 760 nm as a result of the synergistic effects of electron-withdrawing and electron-donating modifications on the p-n-CD electronic structure. The rational p-n junction modification in NIR-phosphorescent p-n-CDs creates the narrow bandgap and stabilize triplet states. Importantly, the p-n-CDs exhibit a phosphorescent efficiency as high as 17.6% in aqueous solutions and high NIR emission brightness comparable to that of the clinically used ICG. These RTP properties render p-n-CDs as a superior long-lived NIR imaging platform for biomedical applications with minimal background interference.

2. The sonodynamic mechanism of p-n-CDs needs more careful investigation and explanation.

It is not clear why p-n-CDs can generate ROS under US.

Reply: The sonodynamic mechanism of p-n-CDs was discussed in the Discussion section.

Long-lived triplet states of organic photosensitizers are tremendously important for photodynamic therapy (PDT) owing to a high yield of singlet oxygen generation upon photo-excitation. Based on

similar mechanisms to organic photosensitizers, organic sonosensitizers with long-lived triplet states are also favorable for SDT under US irradiation. Interestingly, phosphorescent p-n-CDs have long-lived triplet states in aqueous solutions, enabling effective singlet oxygen generation for SDT like organic photosensitizers. Unlike organic sonosensitizers with low photostability, p-n-CDs have excellent photostability owing to the graphitic structure. Moreover, p-n-CDs possess a narrow bandgap for enhanced sonoexcitation.

3. What is the stability and blood circulation behavior of p-n-CDs without any modification in physiological conditions? Please add the related data.

Reply: Like p-n-CD@CCM, free p-n-CDs were also kept stable in physiological conditions (PBS and fetal bovine serum) because of high water solubility (Supplementary Fig. 8)

Supplementary Fig. 26 showed the plasma concentration-time profile of intravenously injected p-n-CDs. For free p-n-CDs without CCM coating, the elimination half-life ($t_{1/2}$) was only 0.8 ± 0.1 h, revealing rapid renal clearance owing to the ultrafine size (2.2 ± 0.4 nm), like previous reports on CDs.

We have added these results and corresponding discussion in the revised manuscript.

Supplementary Fig. 8 Photographs of p-n-CD aqueous solution, PBS solution, and fetal bovine serum (FBS) solution stored for different periods of time (0, 1, and 7 days).

Supplementary Fig. 26 Plasma concentration-time profiles (a) and pharmacokinetic parameters (b) of p-n-CD and p-n-CD@CCM after intravenous administration of various drug formulations at the p-n-CD dose of 2.0 mg/kg (n=5).

4. The antitumor mechanism of p-n-CDs in cellular level should be explored.

Reply: To investigate the antitumor mechanism of p-n-CDs in cellular level, we evaluated the intracellular ROS levels using DCFH-DA staining assay. The 143B cells in the control group, US irradiation alone group, and p-n-CD@CCM alone group exhibited no obvious ROS fluorescence signal. In contrast, the strongest green fluorescence located in the cytoplasm was observed in p-n-CD+US and p-n-CD@CCM+US groups (Fig. 51 and Supplementary Fig. 18, 19b), confirming the ROS-mediated cell death induced by p-n-CD-based SDT.

5. The results in Figure 6b and 6c are the in vivo NIR imaging of mice, which shows that there was obvious fluorescence signal in the liver at 0 h post-injection, why?

Reply: As the reviewer said, fluorescent signal excited at 700 nm was observed in the abdomen of the mice at 0 h post-injection, which could be ascribed to the fluorescent emission from the food in the intestinal tract rather than from p-n-CD@CCM.

6. In the in vivo phase, the data shows only one round of treatment is enough to inhibit tumor growth. Whether tumor cells in tissue were apoptosis on day 4 or 5, or other long-term antitumor mechanism? Please explain it with data.

Reply: To investigate whether tumor cells in tissue were apoptosis on day 4 or 5, the tumor sections in each groups were collected on the fourth day post treatment and subjected to terminal deoxynucleotidyl transferase dUTP nick-end labeling (TUNEL) staining. As shown in Supplementary

Fig. 25, the tumor in p-n-CD@CCM + US group showed severe apoptosis and necrosis.

We have added these results and corresponding discussion in the revised manuscript.

Supplementary Fig. 25. TUNEL staining of the tumors after different treatments.

7. The legend in Figure 6g and 6h should be checked again.

Reply: Thanks for your careful work. We have modified the legend in Fig. 6g and 6h in the revised manuscript.

Fig. 6 (g) Survival curves of mice after indicated treatments. (h) Body weight of mice after indicated treatments.

REVIEWER COMMENTS

Reviewer #1 (Remarks to the Author):

The manuscript explored a NIR-emitting sonosensitizers from a NIR phosphorescent carbon dot (CD) material to enhance SDT as thermodynamically and dynamically favorable factors under low-intensity ultrasound irradiation. From an innovative perspective, the reviewer doesn't think the manuscript can be accepted by Nature Communications. Further, the reviewer really doubts the authenticity of the data provided by the authors, such as TEM images of p-n-CD@CCM in Fig. 5a and Supplementary Fig. 14, and the quantitative results of live/dead cell staining in Fig 5k and Supplementary Fig. 19. Below are some comments for the authors' consideration.

1. This manuscript was lack of innovation. There were so many reports about cancer cell camouflaged nanoparticles to target tumor sites: *Adv. Mater.* 28, 3460-3466 (2016). *Journal of nanobiotechnology* 18, 60 (2020). *ACS Nano* 13, 2849-2857 (2019). As for the innovation of this manuscript, the reviewer doesn't think it can be published in Nature Communications.
2. TEM images in Fig.5a showed that p-n-CD@CCM with dark p-n-CDs, but TEM images of p-n-CD@CCM were all dark in Supplementary Fig. 14 provided by the authors. They did not have similar appearance. How could the authors judge that the nanoparticles in Supplementary Fig. 14 were p-n-CD@CCM?
3. The TEM images of p-n-CD@CCM in Fig. 5a was larger than 100 nm, but particle size in Supplementary Fig. 14 were ~70 nm. They did not even have similar particle size. Which one was wrong? I deeply doubt the authenticity of the data provided by the authors.
4. The authors know that suitable particle size is a key factor that is advantageous to tumor tissue accumulation and tumor penetration. Further, the cancer targeting ability, immune escape ability and tumor penetration ability are also affected by the nanoparticle size. The manuscript did not reflect the necessity of selecting 78 nm as the particle size of nanopatform. Why no nanoparticles with different particle sizes were compared with the cancer targeting ability, immune escape ability and tumor penetration ability? Would nanoparticles with other particle sizes had a higher cancer targeting ability, immune escape ability and tumor penetration ability?
5. In the preparation of p-n-CD@CCM, 1 mL p-n-CDs (0.3 mg/mL) were mixed with 1 mL CCM dispersion. Why this proportion of p-n-CDs and CCM was selected? Would the proportion affect the cancer targeting ability and tumor therapeutic effect of nanoparticles?
6. The authors provided the quantitative analysis of live/dead cell staining images of Fig. 5k in Supplementary Fig. 19, but the proportion of living or dead cells was not provided. Furthermore, why the cells in ICG + US group had same green intensity with control group and same red intensity with p-n-CD + US group? The reviewer doubts the consistency and authenticity of the data in Fig. 5k, Supplementary Fig. 17, and Supplementary Fig. 19.
7. The apoptosis of 143B cells after indicated treatments should examined by flow cytometry.

Reviewer #3 (Remarks to the Author):

In this work, the authors report the first phosphorescent sonosensitizer concept as a new class of theranostic platforms for NIR imaging guided SDT. Unlike previous reports on nanoparticle sonosensitizer engineering with a narrowed bandgap from a thermodynamic perspective, this work reports a novel thermodynamic/dynamical engineering route to SDT

sonosensitizers from NIR phosphorescent carbon dot (CD) material with a narrow bandgap and long-lived excited triplet states. Some important pharmaceuticals data and the detailed discussion of the NIR phosphorescent mechanisms have been added in the revised manuscript. Overall, this work is interesting and well organized, which is of importance in development of CD-based sonosensitizers and NIR probes for theranostic applications. Therefore, this manuscript should be considered for publication in Nature Communications after a minor revision.

1. Owing to the presence of long-lived triplet excited states, the NIR-phosphorescent p-n-CDs exhibit outstanding sonodynamic performance. How about the photodynamic properties of p-n-CDs under NIR laser irradiation?
2. After CCM encapsulation, the sonodynamic performance of p-n-CD@CCM should be measured.
3. The authors claim that the US irradiation possesses higher tissue penetration deepness. What's the distance of the US probe from the tumor during the US treatment? Thus, please add a deep site tumor model to confirm the efficient SDT performance of p-n-CD@CCM.
4. In Fig. 4I, the ESR spectrum of only US treatment should be provided.
5. During the SDT performance measurements, the concentration of the samples should be given.

Itemized Responses to Reviewers' Comments

Reviewer #1

The manuscript explored a NIR-emitting sonosensitizers from a NIR phosphorescent carbon dot (CD) material to enhance SDT as thermodynamically and dynamically favorable factors under low-intensity ultrasound irradiation. From an innovative perspective, the reviewer doesn't think the manuscript can be accepted by Nature Communications. Further, the reviewer really doubts the authenticity of the data provided by the authors, such as TEM images of p-n-CD@CCM in Fig. 5a and Supplementary Fig. 14, and the quantitative results of live/dead cell staining in Fig 5k and Supplementary Fig. 19. Below are some comments for the authors' consideration.

Reply: Although some comments are relatively single-faceted, we thank the reviewer for these suggestions for improving our manuscript. We have addressed all of Reviewer #1's comments and concerns in detail, and the corresponding revisions have been made in the revised manuscript and supporting information. Our responses to these issues are attached as follows.

1. **This manuscript was lack of innovation. There were so many reports about cancer cell camouflaged nanoparticles to target tumor sites: Adv. Mater. 28, 3460-3466 (2016). Journal of nanobiotechnology 18, 60 (2020). ACS Nano 13, 2849-2857 (2019). As for the innovation of this manuscript, the reviewer doesn't think it can be published in Nature Communications.**

Reply: We noted that Reviewer #1 in the first round of review thought our manuscript lacked necessary data (e.g. physiological stability and pharmacokinetic data) rather than innovation. Surprisingly, in the second round of review, Reviewer #1 rejected our manuscript just because **there were so many reports about cancer cell camouflaged nanoparticles to target tumor sites**. We can't understand the reason why Reviewer #1 overlooked our significant findings in phosphorescent CD sonosensitizers as a new class of theranostic platforms for NIR imaging guided sonodynamic therapy. We would not accept the partial argument on the insufficient innovation of p-n-CD@CCM, in which CCM was just utilized as effective drug carrier and targeting agent in the structural design. Reviewer #1 should judge whether our underlined **NIR phosphorescent sonosensitizers** as theranostic platforms are novel, as did other reviewers (Reviewer #1 and # 3). If such phosphorescent sonosensitizers were reported

previously, the reason for rejection would be persuasive. However, in the first and second rounds of review, Reviewer #1 had no queries about innovation of the most crucial ingredient (p-n-CD), which was elaborately designed as a novel NIR phosphorescent probe and a highly efficient sonosensitizer via fine surface modification with novel p-n junctions. We believe that our important findings on CD-based sonosensitizers and NIR probes can meet the high standards of *Nature Communications*.

2. TEM images in Fig. 5a showed that p-n-CD@CCM with dark p-n-CDs, but TEM images of p-n-CD@CCM were all dark in Supplementary Fig. 14 provided by the authors. They did not have similar appearance. How could the authors judge that the nanoparticles in Supplementary Fig. 14 were p-n-CD@CCM?

Reply: In the last version, the single-particle p-n-CD@CCM (Fig. 5a) was imaged by JEM-2100F (high-resolution mode at 200 kV). In contrast, the broad-view TEM image of p-n-CD@CCM nanoparticles shown in Supplementary Fig. 14 was taken by JEM-1400 Flash (120 kV) at low resolution, so p-n-CD@CCM nanoparticles appeared at all dark contrast. If we simply zoom in the low-resolution TEM image, you will see similar appearance with pale CCM edge and dark CD core, as shown in the following.

In the current version, we replaced the low-resolution, wide-view TEM image with the high-resolution, wide-view TEM image taken at 200 kV by JEM-2100F. As shown in Fig. 5a, all particles with similar CD contrast to that of the single-particle p-n-CD@CCM (Fig. 5b) were observed consistently. The HRTEM image was re-arranged in Supplementary Fig. 15 instead.

The zoomed in low-resolution TEM image of p-n-CD@CCM nanoparticles taken at 120 kV.

Fig. 5 (a, b) TEM images of p-n-CD@CCM

3. The TEM images of p-n-CD@CCM in Fig. 5a was larger than 100 nm, but particle size in Supplementary Fig. 14 were ~70 nm. They did not even have similar particle size. Which one was wrong? I deeply doubt the authenticity of the data provided by the authors.

Reply: First, Reviewer #1 should note that a wide distribution in size is common for nanoparticles, including CCM carriers and their nanodrugs. It is also noted that TEM images only show limited sample areas not all areas. The wide-view TEM image (Fig. 5a) does show that most p-n-CD@CCM particles have size smaller than 100 nm. Single particle TEM image (Fig. 5b) and wide-view TEM image show consistent sample size information. Importantly, we also measured the size distribution of p-n-CD@CCM by DLS. As shown in Fig. 5d, the hydrodynamic average diameter of p-n-CD@CCM is 78.8 ± 1.3 nm and there is about 22.8% of nanoparticles have size larger than 100 nm. All these measurements show consistent size distribution without doubt.

Fig. 5 (d) DLS size distribution of free p-n-CDs, CCM, and p-n-CD@CCM.

4. The authors know that suitable particle size is a key factor that is advantageous to tumor tissue accumulation and tumor penetration. Further, the cancer targeting ability, immune escape ability and tumor penetration ability are also affected by the nanoparticle size. The manuscript did not reflect the necessity of selecting 78 nm as the particle size of nanopatform. Why no nanoparticles with different particle sizes were compared with the cancer targeting ability, immune escape ability and tumor penetration ability? Would nanoparticles with other particle sizes had a higher cancer targeting ability, immune escape ability and tumor penetration ability?

Reply: Generally, the intratumor accumulation levels of common nanoparticles without cancer-specific targeting agents are greatly affected by nanoparticle size (the well-known EPR effect). In those

cases, the size effects of nanoparticles on intratumor accumulation should be explored carefully. However, for nanoparticles coated with cancer-specific targeting agents such as CCM, the most dominant targeting factor is cancer-specific molecules rather than particle size. Because of this specificity of CCM, the insignificant size-targeting relationship was neglected. Our goal is to develop NIR-phosphorescent materials as long-lived probes and enhanced sonosensitizers to achieve the best therapeutic effect of SDT rather than to investigate the effect of nanoparticle size on their tumor-targeting ability.

5. In the preparation of p-n-CD@CCM, 1 mL p-n-CDs (0.3 mg/mL) were mixed with 1 mL CCM dispersion. Why this proportion of p-n-CDs and CCM was selected? Would the proportion affect the cancer targeting ability and tumor therapeutic effect of nanoparticles?

Reply: The experimental section in the preparation of p-n-CD@CCM was revised with more details added (See Methods). Briefly, 1 mL aqueous solution of p-n-CDs with varied concentration (0.1, 0.2, 0.3, 0.4, or 0.5 mg/mL) was mixed with CCM-143B or CCM-Hela dispersions (1 mL, 1.0 mg/mL). The mixture was transferred into a syringe and successively extruded through 500 nm, 200 nm, and 100 nm water-phase filters, followed by centrifugation to remove unloaded p-n-CDs from the dispersions. The amount of encapsulated p-n-CDs in p-n-CD@CCM was determined by UV-vis-NIR absorption spectroscopy. Finally, a p-n-CD@CCM dispersion with encapsulated p-n-CDs at a high concentration (200 $\mu\text{g/mL}$) and a high loading ($\sim 20\%$) was obtained at the mass ratio of p-n-CD to CCM (3:10 w/w) for further structural characterization and SDT measurements.

We found that at the mass ratio of p-n-CD to CCM (3:10 w/w), p-n-CDs were encapsulated in CCM carriers at a high concentration (200 $\mu\text{g/mL}$) and a high loading ($\sim 20\%$). This is why we selected the mixing ratio. Generally, the targeting ability of cancer cell camouflaged nanoparticles is dependent on cancer-specific proteins retained on the surface of CCM, such as pan-cadherin, Na^+/K^+ -ATPase, and gp100. Like many previous reports (Nano Lett., 2014, 14, 2181-2188; Nano Lett., 2016, 16, 5895-5901; ACS Nano, 2018, 12, 1350-1358), p-n-CD@CCM-143B and p-n-CD@CCM-Hela were also equipped with the targeting capability from CCM specific proteins. We think that the mass ratio of p-n-CD to CCM has little effects on cancer targeting.

6. The authors provided the quantitative analysis of live/dead cell staining images of Fig. 5k in Supplementary Fig. 19, but the proportion of living or dead cells was not provided. Furthermore, why the cells in ICG + US group had same green intensity with control group and same red intensity with p-n-CD + US group? The reviewer doubts the consistency and authenticity of the data in Fig. 5k, Supplementary Fig. 17, and Supplementary Fig. 19.

Reply: According to the reviewer's suggestion, we have provided the proportion of live cells or dead cells based on the live/dead cell staining images (Supplementary Fig. 21a). The semi-quantitative analysis was determined by the Image J software via triplicate parallel samples.

Indeed, the ICG + US group should have lower green intensity than control group and lower red intensity than p-n-CD + US group under the same cell total (live + dead cells). The similar green/red intensity was observed likely because the ICG + US group had a larger cell total than other groups, as shown in Supplementary Fig. 19 (Merged).

Generally, the semi-quantitative analysis of above-mentioned mean fluorescence intensity may make misunderstandings in some cases such as significant difference in cell total. To accurately reflect the SDT effect on the cancer cell apoptosis at the cell level, we further provided the proportions of live and dead cells according to the reviewer's suggestion (Supplementary Fig. 21a). Thus, the consistency and authenticity of the data can be confirmed.

Supplementary Fig. 21 (a) Semi-quantitative analysis of live/dead cell staining determined by the Image J software via triplicate parallel samples, indicating the proportion of live cells or dead cells in different groups.

7. The apoptosis of 143B cells after indicated treatments should be examined by flow cytometry.

Reply: According to the reviewer's suggestion, we performed the apoptosis assay through the flow cytometry using an Annexin V-FITC/PI assay kit according to the manufacturer's protocol. As presented in Supplementary Fig. 22, significant apoptosis of 143B cells was detected in the p-n-CD+US and p-n-CD@CCM+US groups, which was consistent with the live/dead cell staining results. We have added these results and corresponding discussion in the revised manuscript.

Supplementary Fig. 22 Flow cytometry apoptosis assay of 143B cells after different treatments.

Reviewer #3

In this work, the authors report the first phosphorescent sonosensitizer concept as a new class of theranostic platforms for NIR imaging guided SDT. Unlike previous reports on nanoparticle sonosensitizer engineering with a narrowed bandgap from a thermodynamic perspective, this work reports a novel thermodynamic/dynamical engineering route to SDT sonosensitizers from NIR phosphorescent carbon dot (CD) material with a narrow bandgap and long-lived excited triplet states. Some important pharmaceuticals data and the detailed discussion of the NIR phosphorescent mechanisms have been added in the revised manuscript. Overall, this work is interesting and well organized, which is of importance in development of CD-based

sonosensitizers and NIR probes for theranostic applications. Therefore, this manuscript should be considered for publication in Nature Communications after a minor revision.

Reply: Thanks for the reviewer's careful, valuable suggestions, and positive evaluation. We have addressed all of Reviewer #3's comments and concerns in detail, and the corresponding revisions have been made in the revised manuscript and supporting information. Our responses to these issues are attached as follows.

1. Owing to the presence of long-lived triplet excited states, the NIR-phosphorescent p-n-CDs exhibit outstanding sonodynamic performance. How about the photodynamic properties of p-n-CDs under NIR laser irradiation?

Reply: According to the reviewer's suggestion, we have investigated the photodynamic properties of p-n-CDs under 660 nm laser irradiation using DPBF as the $^1\text{O}_2$ probe. As depicted in Supplementary Fig. 14, the characteristic absorption peak of the $^1\text{O}_2$ probe DPBF at 412 nm decreased significantly with p-n-CD after 660 nm laser irradiation for 10 min, confirming the excellent photodynamic properties of p-n-CDs owing to the presence of long-lived triplet excited states. We have added these results and corresponding discussion in the revised manuscript.

Supplementary Fig. 14 Time-dependent $^1\text{O}_2$ generation of p-n-CDs under 660 nm laser irradiation (0.2 W cm^{-2}) for 10 min detected with the DPBF probe.

2. After CCM encapsulation, the sonodynamic performance of p-n-CD@CCM should be measured.

Reply: Thanks for your suggestion. We have investigated the sonodynamic properties of p-n-CD@CCM under a low-intensity US irradiation (50 kHz , 3.0 W cm^{-2}) using the DPBF probe. The

characteristic absorption peaks of DPBF at 412 nm rapidly disappeared for p-n-CD@CCM with increasing the US irradiation time (Supplementary Fig. 18a). The $^1\text{O}_2$ generation efficiency of p-n-CD@CCM was thus calculated to be 0.212 min^{-1} (Supplementary Fig. 18b), which was similar to that of pristine p-n-CD (0.220 min^{-1}), indicating that encapsulation into CCM will not affect the excellent sonodynamic performance of p-n-CDs. We have added these results and corresponding discussion in the revised manuscript.

Supplementary Fig. 18 (a) Time-dependent $^1\text{O}_2$ generation of p-n-CD@CCM under US irradiation (50 kHz, 3.0 W cm^{-2}) for 5 min detected with the DPBF probe. (b) Rate constant of US triggered $^1\text{O}_2$ generation in the presence of p-n-CDs and p-n-CD@CCM under the same US irradiation.

3. The authors claim that the US irradiation possesses higher tissue penetration deepness.

What's the distance of the US probe from the tumor during the US treatment? Thus, please add a deep site tumor model to confirm the efficient SDT performance of p-n-CD@CCM.

Reply: According to the reviewer's suggestion, we evaluated the potential therapeutic effect of p-n-CD@CCM for *in vivo* deep-tissue SDT. As illustrated in Supplementary Fig. 29a, two 143B tumors were established on the left and right sides of mice and the US irradiation was carried out from left to the right. The left and right tumors rapidly grew in the control group (Supplementary Fig. 29b). Significantly, the tumor growth on both sides was completely inhibited for mice treatment with p-n-CD@CCM plus US irradiation from left to right. These results clearly demonstrated that the US irradiation could penetrate through the whole body of mice and the high-efficiency p-n-CD@CCM sonosensitizers could be applicable for deep-tissue sonodynamic cancer therapy. We have added these results and corresponding discussion in the revised manuscript.

Supplementary Fig. 29 (a) Schematic illustration of p-n-CD@CCM treatments to 143B tumors in the left side and right side via the SDT. The US penetrated from left to right. (b) 143B tumor growth curves of mice after indicated treatments.

4. In Fig. 4l, the ESR spectrum of only US treatment should be provided.

Reply: Thanks for your suggestion. We have added the ESR spectrum of only US treatment in the revised manuscript.

Fig. 4 (l) ESR spectra of US triggered ¹O₂ generation in the presence of p-n-CDs, p-CDs, n-CDs, ICG, or TiO₂ at the same concentrations (200 µg/mL) using TEMP as the trapping agent of ¹O₂. The US irradiation alone was also detected.

5. During the SDT performance measurements, the concentration of the samples should be given.

Reply: Thanks for your suggestion. The concentration of p-n-CD, p-CD, n-CD, ICG, and TiO₂ during the SDT performance measurements was all 200 µg/mL. We have added these results in the legend of Fig. 4.

Itemized Responses to Reviewers' Comments

Reviewer #1

The manuscript explored a NIR-emitting sonosensitizers from a NIR phosphorescent carbon dot (CD) material to enhance SDT as thermodynamically and dynamically favorable factors under low-intensity ultrasound irradiation. From an innovative perspective, the reviewer doesn't think the manuscript can be accepted by Nature Communications. Further, the reviewer really doubts the authenticity of the data provided by the authors, such as TEM images of p-n-CD@CCM in Fig. 5a and Supplementary Fig. 14, and the quantitative results of live/dead cell staining in Fig 5k and Supplementary Fig. 19. Below are some comments for the authors' consideration.

Reply: Many thanks for Reviewer #1's second review with valuable suggestions for improving our manuscript. After reading the comments, we realized that Reviewer #1 has not understood or recognized our advances in the field of NIR phosphorescent sonosensitizers. We have thus to make additional statement that clarifies the novelty of our work. We have revised the introduction section to expound our innovations. We believe that our important findings on CD-based sonosensitizers and NIR probes can meet the high standards of *Nature Communications*. We would be grateful if the Reviewer #1 could re-evaluate our major innovations in the context of NIR-phosphorescent SDT materials rather than cancer cell camouflaged nanoparticles.

1. **This manuscript was lack of innovation. There were so many reports about cancer cell camouflaged nanoparticles to target tumor sites: Adv. Mater. 28, 3460-3466 (2016). Journal of nanobiotechnology 18, 60 (2020). ACS Nano 13, 2849-2857 (2019). As for the innovation of this manuscript, the reviewer doesn't think it can be published in Nature Communications.**

Reply: Indeed, there were so many reports about cancer cell camouflaged nanoparticles to target tumor sites. In our manuscript, the cancer-specific targeting and high-level intratumor enrichment of NIR phosphorescent sonosensitizers were achieved by CCM encapsulation for precision SDT. However, our major innovations are not just the design and construction of sonosensitizer delivery systems by employing cancer cell membrane (CCM) as the cancer-specific targeting agent. Instead, **we report the first 'phosphorescent sonosensitizer' concept as a new class of theranostic platforms for NIR**

imaging guided sonodynamic therapy. Unlike previous reports that only consider the reduction of sonosensitizer bandgap from a thermodynamic perspective, our work reports a novel thermodynamic/dynamical engineering route to SDT sonosensitizers from NIR phosphorescent CD material with a narrow bandgap and long-lived excited triplet states. To clarify the novelty of our work, several important innovations were expounded from points of view in structure, methodology, property/function, mechanism, and application:

- 1) **We report the novel structural design of multifunctional CDs (p-n-CDs) with an ingenious p-n junction and an extremely narrow bandgap** for enhanced charge separation dynamics and NIR excitation/emission. These distinct electronic structures endowed CDs with carrier excitation at low energy, charge separation at high efficiency, and great potential for many modern technologies, including LEDs, solar cells, photocatalytic agents, bioimaging, SDT, and PDT.
- 2) **We have developed a one-step synthetic strategy (microwave synthesis) for controlled synthesis of p-n junction CDs.** Moreover, we can facilely tune the conductivity types of CDs (p-, n-, and p-n junction) by surface engineering with sulfonic acceptors and/or N donators. Compared with previously reported preparation of p-n junction CDs involving complex post-treatments and long-time consumption (Adv. Mater., 2014, 26, 3297-3303), our one-step and high-efficiency method is suitable for low-cost production.
- 3) **Interestingly, we found that the p-n junction CDs possessed bright long-lived (11.4 μ s) phosphorescence rather than widely reported short-lived (ns) fluorescence from CDs or graphene oxide quantum dots** (Adv. Mater., 2014, 26, 3297-3303; ACS Nano, 2018, 12, 3523-3532). Although they were dispersed in aqueous solutions or physiological media, they exhibited afterglow emission at room temperature without the need of stabilizing agents such as PVA or other matrices. It is noted that current afterglow CDs in the hybrid systems are mainly limited to visible RTP emissions from blue, green to yellow. **Here, we found that p-n-CDs can emit NIR phosphorescence at 760 nm with phosphorescent efficiency as high as 17.6%** as a result of the synergistic effects of electron-withdrawing and electron-donating modifications on the electronic structure.
- 4) **We also found that the p-n junction CDs exhibited enhanced $^1\text{O}_2$ generation efficiency under**

US irradiation through three enhancement mechanisms: (1) highly effective inhibition of the e^-h^+ pair recombination through the p-n junction, (2) long-lived triplet state mediated 1O_2 generation, (3) GSH depletion using overexpressed GSH as a hole sacrificial agent.

- 5) **We constructed p-n-CD@CCM vehicles with enhanced cancer-specific targeting and high-level intratumor enrichment** using CCM as the targeting agent and delivery system to overcome the in vivo limitations by free-standing CDs, such as rapid renal clearance owing to the ultrafine size.
 - 6) Given the NIR imaging capability, excellent biocompatibility, enhanced US-mediated 1O_2 generation efficiency of p-n-CDs as well as the CCM targeting capability, **we finally constructed a novel CD-based theranostic platform (p-n-CD@CCM) for NIR imaging guided sonodynamic therapy with excellent tumor eradication capability.**
 - 7) **Our results will open up a new approach to engineer phosphorescent materials with NIR emission and long-lived triplet excited states for wide-range applications** apart from sonodynamic precision tumor therapy.
2. **TEM images in Fig. 5a showed that p-n-CD@CCM with dark p-n-CDs, but TEM images of p-n-CD@CCM were all dark in Supplementary Fig. 14 provided by the authors. They did not have similar appearance. How could the authors judge that the nanoparticles in Supplementary Fig. 14 were p-n-CD@CCM?**

Reply: In the last version, the single-particle p-n-CD@CCM (Fig. 5a) was imaged by JEM-2100F (high-resolution mode at 200 kV). In contrast, the broad-view TEM image of p-n-CD@CCM nanoparticles shown in Supplementary Fig. 14 was taken by JEM-1400 Flash (120 kV) at low resolution, so p-n-CD@CCM nanoparticles appeared at all dark contrast. If we simply zoom in the low-resolution TEM image, you will see similar appearance with pale CCM edge and dark CD core, as shown in the following.

In the current version, we replaced the low-resolution, wide-view TEM image with the high-resolution, wide-view TEM image taken at 200 kV by JEM-2100F. As shown in Fig. 5a, all particles with similar CD contrast to that of the single-particle p-n-CD@CCM (Fig. 5b) were observed consistently. The HRTEM image was re-arranged in Supplementary Fig. 15 instead.

The zoomed in low-resolution TEM image of p-n-CD@CCM nanoparticles taken at 120 kV.

Fig. 5 (a, b) TEM images of p-n-CD@CCM

3. The TEM images of p-n-CD@CCM in Fig. 5a was larger than 100 nm, but particle size in Supplementary Fig. 14 were ~70 nm. They did not even have similar particle size. Which one was wrong? I deeply doubt the authenticity of the data provided by the authors.

Reply: First, a wide distribution in size is common for nanoparticles, including CCM carriers and their nanodrugs (ACS Nano, 2018, 12, 8520-8530; Biomaterials, 2020, 257, 120256; Nano Lett., 2014, 14, 2181-2188). Moreover, TEM images only show limited sample areas not all areas. Compared to TEM characterization, DLS can measure the size distribution of the sample, which is more statistically significant. The DLS results revealed that the number of p-n-CD@CCM with a size of 78.8 nm was the most (Fig. 5d), but there is about 22.8% of nanoparticles have size larger than 100 nm. In the current version, we replaced the low-resolution, wide-view TEM image with the high-resolution, wide-view TEM image taken at 200 kV by JEM-2100F. The wide-view TEM image (Fig. 5a) and single particle TEM image (Fig. 5b) show consistent sample size information. Therefore, the data integrity and consistency can be confirmed by the TEM and DLS results.

Fig. 5 (a, b) TEM images of p-n-CD@CCM

Fig. 5 (d) DLS size distribution of free p-n-CDs, CCM, and p-n-CD@CCM.

4. The authors know that suitable particle size is a key factor that is advantageous to tumor tissue accumulation and tumor penetration. Further, the cancer targeting ability, immune escape ability and tumor penetration ability are also affected by the nanoparticle size. The manuscript did not reflect the necessity of selecting 78 nm as the particle size of nanopatform. Why no nanoparticles with different particle sizes were compared with the cancer targeting ability, immune escape ability and tumor penetration ability? Would nanoparticles with other particle sizes had a higher cancer targeting ability, immune escape ability and tumor penetration ability?

Reply: Generally, the intratumor accumulation levels of common nanoparticles without cancer-specific targeting agents are greatly affected by nanoparticle size (the well-known EPR effect). In those cases, the size effects of nanoparticles on intratumor accumulation should be explored carefully. For example, particles with a diameter less than 400 nm can extravasate from leaky vasculature into tumor interstitium (Nat. Nanotechnol., 2007, 2, 751). In addition, to reduce liver capture and renal filtration, the generally accepted size of the nanoparticles is in the range of 20-200 nm (J. Controlled Release, 2010, 148, 135; J. Am. Chem. Soc., 2013, 135, 4978-4981). However, for nanoparticles coated with cancer-specific targeting agents such as CCM, the most dominant targeting factor is cancer-specific molecules rather than particle size. CCM derived from homologous tumors have been developed as novel tumor-specific delivery vehicles because cancer-specific proteins are retained on the surface enabling homotypic recognition to the same cancer cell lines (Nano Lett., 2014, 14, 2181-2188; ACS Nano, 2018, 12, 1350-1358). Moreover, CCM coated nanoparticles are capable of immune escape capability, which rely on the surface membrane proteins of cancer cells (Nano Lett., 2016, 16, 5895-5901; ACS Nano 2021, 15, 19756-19770; Adv. Mater. 2016, 28, 3460-3466). Encouraged by the immune escaping and homologous binding capabilities of CCM, the cancer-specific targeting and high-level intratumor enrichment of theranostic CDs were achieved for precision SDT with complete eradication of solid tumors by single injection and single irradiation (Fig. 6).

5. In the preparation of p-n-CD@CCM, 1 mL p-n-CDs (0.3 mg/mL) were mixed with 1 mL CCM dispersion. Why this proportion of p-n-CDs and CCM was selected? Would the proportion affect the cancer targeting ability and tumor therapeutic effect of nanoparticles?

Reply: The experimental section in the preparation of p-n-CD@CCM was revised with more details added (See Methods). Briefly, 1 mL aqueous solution of p-n-CDs with varied concentration (0.1, 0.2, 0.3, 0.4, or 0.5 mg/mL) was mixed with CCM-143B or CCM-Hela dispersions (1 mL, 1.0 mg/mL). The mixture was transferred into a syringe and successively extruded through 500 nm, 200 nm, and 100 nm water-phase filters, followed by centrifugation to remove unloaded p-n-CDs from the dispersions. The amount of encapsulated p-n-CDs in p-n-CD@CCM was determined by UV-vis-NIR absorption spectroscopy. Finally, a p-n-CD@CCM dispersion with encapsulated p-n-CDs at a high concentration (200 μ g/mL) and a high loading (~20%) was obtained at the mass ratio of p-n-CD to CCM (3:10 w/w) for further structural characterization and SDT measurements.

We found that at the mass ratio of p-n-CD to CCM (3:10 w/w), p-n-CDs were encapsulated in CCM carriers at a high concentration (200 μ g/mL) and a high loading (~20%). This is why we selected the mixing ratio. Generally, the targeting ability of cancer cell camouflaged nanoparticles is dependent on cancer-specific proteins retained on the surface of CCM, such as pan-cadherin, Na⁺/K⁺-ATPase, and gp100. Like many previous reports (Nano Lett., 2014, 14, 2181-2188; Nano Lett., 2016, 16, 5895-5901; ACS Nano, 2018, 12, 1350-1358), p-n-CD@CCM-143B and p-n-CD@CCM-Hela were also equipped with the targeting capability from CCM specific proteins. We think that the mass ratio of p-n-CD to CCM has little effects on cancer targeting.

6. The authors provided the quantitative analysis of live/dead cell staining images of Fig. 5k in Supplementary Fig. 19, but the proportion of living or dead cells was not provided. Furthermore, why the cells in ICG + US group had same green intensity with control group and same red intensity with p-n-CD + US group? The reviewer doubts the consistency and authenticity of the data in Fig. 5k, Supplementary Fig. 17, and Supplementary Fig. 19.

Reply: According to the reviewer's suggestion, we have provided the proportion of live cells or dead cells based on the live/dead cell staining images (Supplementary Fig. 21a). The semi-quantitative analysis was determined by the Image J software via triplicate parallel samples.

Indeed, the ICG + US group should have lower green intensity than control group and lower red intensity than p-n-CD + US group under the same cell total (live + dead cells). The similar green/red intensity was observed likely because the ICG + US group had a larger cell total than other groups, as

shown in Supplementary Fig. 19 (Merged).

Generally, the semi-quantitative analysis of above-mentioned mean fluorescence intensity may make misunderstandings in some cases such as significant difference in cell total. To accurately reflect the SDT effect on the cancer cell apoptosis at the cell level, we further provided the proportions of live and dead cells according to the reviewer's suggestion (Supplementary Fig. 21a). Thus, the consistency and authenticity of the data can be confirmed.

Supplementary Fig. 21 (a) Semi-quantitative analysis of live/dead cell staining determined by the Image J software via triplicate parallel samples, indicating the proportion of live cells or dead cells in different groups.

7. The apoptosis of 143B cells after indicated treatments should be examined by flow cytometry.

Reply: According to the reviewer's suggestion, we performed the apoptosis assay through the flow cytometry using an Annexin V-FITC/PI assay kit according to the manufacturer's protocol. As presented in Supplementary Fig. 22, significant apoptosis of 143B cells was detected in the p-n-CD+US and p-n-CD@CCM+US groups, which was consistent with the live/dead cell staining results. We have added these results and corresponding discussion in the revised manuscript.

Supplementary Fig. 22 Flow cytometry apoptosis assay of 143B cells after different treatments.

Reviewer #3

In this work, the authors report the first phosphorescent sonosensitizer concept as a new class of theranostic platforms for NIR imaging guided SDT. Unlike previous reports on nanoparticle sonosensitizer engineering with a narrowed bandgap from a thermodynamic perspective, this work reports a novel thermodynamic/dynamical engineering route to SDT sonosensitizers from NIR phosphorescent carbon dot (CD) material with a narrow bandgap and long-lived excited triplet states. Some important pharmaceuticals data and the detailed discussion of the NIR phosphorescent mechanisms have been added in the revised manuscript. Overall, this work is interesting and well organized, which is of importance in development of CD-based sonosensitizers and NIR probes for theranostic applications. Therefore, this manuscript should be considered for publication in *Nature Communications* after a minor revision.

Reply: Thanks for the reviewer's careful, valuable suggestions, and positive evaluation. We have addressed all of Reviewer #3's comments and concerns in detail, and the corresponding revisions have been made in the revised manuscript and supporting information. Our responses to these issues are attached as follows.

1. **Owing to the presence of long-lived triplet excited states, the NIR-phosphorescent p-n-CDs exhibit outstanding sonodynamic performance. How about the photodynamic properties of p-n-CDs under NIR laser irradiation?**

Reply: According to the reviewer's suggestion, we have investigated the photodynamic properties of p-n-CDs under 660 nm laser irradiation using DPBF as the $^1\text{O}_2$ probe. As depicted in Supplementary Fig. 14, the characteristic absorption peak of the $^1\text{O}_2$ probe DPBF at 412 nm decreased significantly with p-n-CD after 660 nm laser irradiation for 10 min, confirming the excellent photodynamic properties of p-n-CDs owing to the presence of long-lived triplet excited states. We have added these results and corresponding discussion in the revised manuscript.

Supplementary Fig. 14 Time-dependent $^1\text{O}_2$ generation of p-n-CDs under 660 nm laser irradiation (0.2 W cm^{-2}) for 10 min detected with the DPBF probe.

2. **After CCM encapsulation, the sonodynamic performance of p-n-CD@CCM should be measured.**

Reply: Thanks for your suggestion. We have investigated the sonodynamic properties of p-n-CD@CCM under a low-intensity US irradiation (50 kHz , 3.0 W cm^{-2}) using the DPBF probe. The characteristic absorption peaks of DPBF at 412 nm rapidly disappeared for p-n-CD@CCM with increasing the US irradiation time (Supplementary Fig. 18a). The $^1\text{O}_2$ generation efficiency of p-n-CD@CCM was thus calculated to be 0.212 min^{-1} (Supplementary Fig. 18b), which was similar to that of pristine p-n-CD (0.220 min^{-1}), indicating that encapsulation into CCM will not affect the excellent sonodynamic performance of p-n-CDs. We have added these results and corresponding discussion in the revised manuscript.

Supplementary Fig. 18 (a) Time-dependent $^1\text{O}_2$ generation of p-n-CD@CCM under US irradiation (50 kHz, 3.0 W cm^{-2}) for 5 min detected with the DPBF probe. (b) Rate constant of US triggered $^1\text{O}_2$ generation in the presence of p-n-CDs and p-n-CD@CCM under the same US irradiation.

3. The authors claim that the US irradiation possesses higher tissue penetration deepness. What's the distance of the US probe from the tumor during the US treatment? Thus, please add a deep site tumor model to confirm the efficient SDT performance of p-n-CD@CCM.

Reply: According to the reviewer's suggestion, we evaluated the potential therapeutic effect of p-n-CD@CCM for *in vivo* deep-tissue SDT. As illustrated in Supplementary Fig. 29a, two 143B tumors were established on the left and right sides of mice and the US irradiation was carried out from left to the right. The left and right tumors rapidly grew in the control group (Supplementary Fig. 29b). Significantly, the tumor growth on both sides was completely inhibited for mice treatment with p-n-CD@CCM plus US irradiation from left to right. These results clearly demonstrated that the US irradiation could penetrate through the whole body of mice and the high-efficiency p-n-CD@CCM sonosensitizers could be applicable for deep-tissue sonodynamic cancer therapy. We have added these results and corresponding discussion in the revised manuscript.

Supplementary Fig. 29 (a) Schematic illustration of p-n-CD@CCM treatments to 143B tumors in the left side and right side via the SDT. The US penetrated from left to right. (b) 143B tumor growth curves of mice after indicated treatments.

4. In Fig. 4l, the ESR spectrum of only US treatment should be provided.

Reply: Thanks for your suggestion. We have added the ESR spectrum of only US treatment in the revised manuscript.

Fig. 4 (l) ESR spectra of US triggered ¹O₂ generation in the presence of p-n-CDs, p-CDs, n-CDs, ICG, or TiO₂ at the same concentrations (200 µg/mL) using TEMP as the trapping agent of ¹O₂. The US irradiation alone was also detected.

5. During the SDT performance measurements, the concentration of the samples should be given.

Reply: Thanks for your suggestion. The concentration of p-n-CD, p-CD, n-CD, ICG, and TiO₂ during the SDT performance measurements was all 200 µg/mL. We have added these results in the legend of Fig. 4.

REVIEWERS' COMMENTS

Reviewer #3 (Remarks to the Author):

This work can be published on Nat. Commun. as it is.